# Direct sulfuric acid formation from the gas-phase oxidation of reduced-sulfur compounds

Torsten Berndt [1], Erik H. Hoffmann [1], Andreas Tilgner [1], Frank Stratmann[2] & Hartmut Herrmann [1]

Sulfuric acid represents a fundamental precursor for new nanometre-sized atmospheric aerosol particles. These particles, after subsequent growth, may influence Earth´s radiative forcing directly, or indirectly through affecting the microphysical and radiative properties of clouds. Currently considered formation routes yielding sulfuric acid in the atmosphere are the gas-phase oxidation of $SO_2$ initiated by OH radicals and by Criegee intermediates, the latter being of little relevance. Here we report the observation of immediate sulfuric acid production from the OH reaction of emitted organic reduced-sulfur compounds, which was speculated about in the literature for decades. Key intermediates are the methylsulfonyl radical, $CH_3SO_2$, and, even more interestingly, its corresponding peroxy compound, $CH_3SO_2OO$. Results of modelling for pristine marine conditions show that oxidation of reduced-sulfur compounds could be responsible for up to ~50% of formed gas-phase sulfuric acid in these areas. Our findings provide a more complete understanding of the atmospheric reduced-sulfur oxidation.

Since more than 3 decades, reduced organic sulfur compounds have been recognized as substantial biogenic emissions contributing to Earth´s sulfur cycle. The sulfur cycle is highly relevant for the Earth's climate due to the ability of the sulfur compound's oxidation products, sulfuric acid ($H_2SO_4$) and methane sulfonic acid (MSA, $CH_3SO_3H$), to generate new airborne particles that effectively scatter incoming solar radiation and affect the formation of cloud condensation nuclei (CCN)[1,2]. CCN in turn may have significant influences on the microphysical and radiative properties[3] and lifetime[4] of clouds.

Globally, the most important organic sulfur compound is dimethyl sulfide (DMS, $CH_3SCH_3$) with an annual emission rate of ~30 million metric tons of sulfur, followed by methylthiol (MeSH, $CH_3SH$) and, to a lesser extent, dimethyl disulfide (DMDS, $CH_3SSCH_3$)[5]. A large number of experimental and theoretical studies have been conducted to ascertain their atmospheric degradation pathways, especially for DMS[6–18], representing the data base for atmospheric models[19–22]. The reaction scheme in Fig. 1 summarises the current knowledge on product formation starting from the methylthiyl ($CH_3S$) and methylsulfonyl radical ($CH_3SO_2$), both formed as important intermediates in the gas-phase oxidation of $CH_3SH$, DMS and DMDS. To the best of our knowledge, up to now there is no experimental evidence for the direct gas-phase formation of $H_2SO_4$, other than via $SO_2$ oxidation by OH radicals[23,24] or Criegee intermediates[25,26], although this has been speculated about in the literature for long[27,28] and such pathways have already been implemented in models[19–22].

Here we experimentally demonstrate the direct formation of $H_2SO_4$ from the OH radical-initiated gas-phase oxidation of organic sulfur compounds by its direct mass spectrometric detection in two flow systems[29–31] under atmospheric conditions with residence times of 7.9 and 32 s. Accompanied modelling shows the importance of this direct pathway for the total $H_2SO_4$ formation in the atmosphere.

[1]Atmospheric Chemistry Department (ACD), Leibniz Institute for Tropospheric Research (TROPOS), 04318 Leipzig, Germany. [2]Atmospheric Microphysics Department (AMP), Leibniz Institute for Tropospheric Research (TROPOS), 04318 Leipzig, Germany. ✉e-mail: berndt@tropos.de

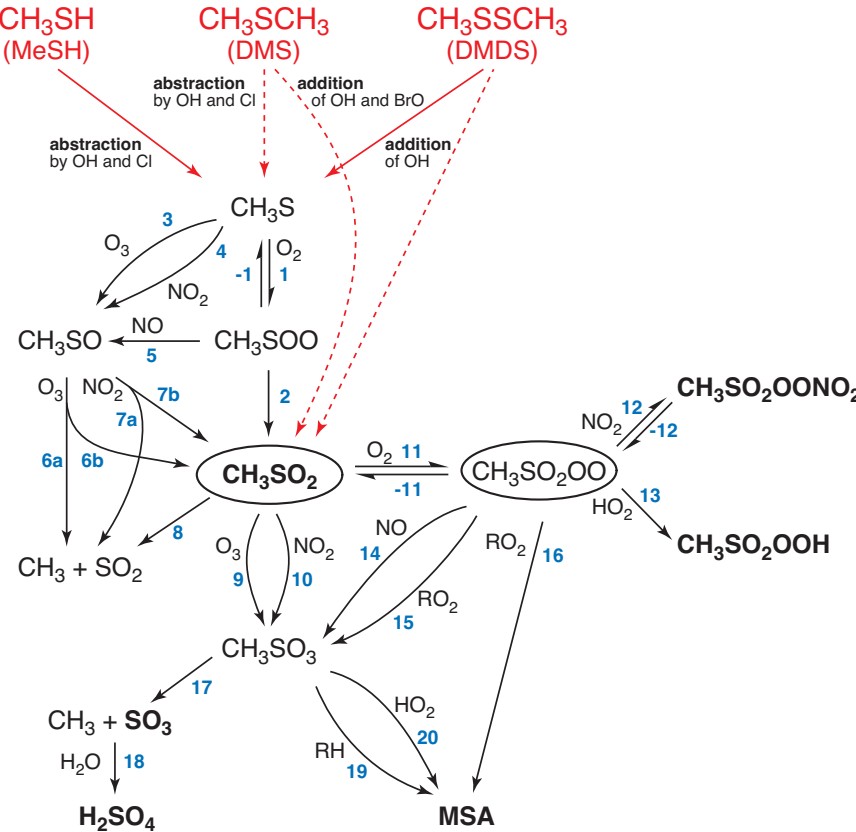

**Fig. 1 | Reaction scheme of the oxidation of reduced-sulfur emissions, i.e. CH₃SH (MeSH, methylthiol), CH₃SCH₃ (DMS, dimethyl sulfide) and CH₃SSCH₃ (DMDS, dimethyl disulfide).** The scheme focuses on the reaction steps relevant for the formation of $H_2SO_4$ and MSA (methane sulfonic acid) starting from $CH_3S$ and $CH_3SO_2$ and is mainly based on Barnes et al.[9]. Signals of observed products in the present study are shown in bold. Dashed red arrows indicate complex reactions to the stated intermediates. Only important main products of the individual pathways are displayed.

## Results and discussion

### Detectable products from CH₃S oxidation

Product ionization by means of iodide (I⁻) and nitrate (NO₃⁻) in the mass spectrometric analysis was found as a suitable way to observe product formation, other than $SO_2$, in the oxidation process. Recently, an experimental $SO_2$ yield of $86 \pm 18\%$ has been reported for low-NO conditions qualifying $SO_2$ as the predominant product[32,33]. Figure 2a shows the detected products, other than $SO_2$, from an overview experiment on the oxidation of $CH_3S$ initiated by the reaction $OH + CH_3SH \rightarrow CH_3S + H_2O$[33,34] (Fig. 1). Product ionisation by iodide allowed to follow $H_2SO_4$ and MSA, signals consistent with the formation of $CH_3SO_2OONO_2$ and $CH_3SO_2OOH$, which were most likely formed via pathways 12 and 13 (Fig. 1), respectively, and the signal of the intermediate $CH_3SO_2$. For the latter, a contribution from $CH_3SOO$ cannot be ruled out. It is to be noted here, that $H_2SO_4$ formation initiated by $OH + SO_2$ is unimportant under the chosen conditions and, thus, $H_2SO_4$ needs to arise from the $CH_3S$ oxidation directly, see also Methods. Comparison of the measured product spectra with results from peak fitting of the spectra supports the signal attribution to the five products identified (Fig. 2b and Supplementary Fig. 1). The signals of all closed-shell products steeply increased with rising $CH_3SH$ conversion while the $CH_3SO_2$ signal levelled off, typical for reactive intermediates. The occurrence of $CH_3SO_2OONO_2$ and $CH_3SO_2OOH$, recently detected from $OH + DMS$ as well[14], indicates $CH_3SO_2OO$ as a significant peroxy species in these reaction systems, which is supported by theoretical calculations[35].

Basically, the direct observation of $CH_3SO_2$ and other intermediates of the $CH_3S$ oxidation for close to atmospheric conditions appears to be very challenging[36,37]. A spectroscopic study on the product formation of $CH_3S + O_2$ in cryogenic matrixes unambiguously identified $CH_3SOO$, $CH_3SO_2$ and $CH_3SO_2OO$ as important intermediates supporting the relevance of the reaction sequence 1/−1, 2 and 11/−11 (Fig. 1) in the $CH_3S$ oxidation[15]. Cryogenic matrix techniques in general represent an useful approach for qualitative studying sulfur oxidation[38].

Formation of $H_2SO_4$ and MSA in our experiments was also observed by means of nitrate ionisation confirming the findings using iodide ionisation (Supplementary Fig. 2).

### MSA formation induced by elevated CH₃SH and DMDS concentrations

It is remarkable that $H_2SO_4$ and MSA concentrations increased almost uniformly with rising $CH_3SH$ conversion, which was accompanied by a rising $HO_2$ radical level for the chosen reaction conditions (Fig. 2a and Supplementary Fig. 2). The competing steps 17 vs. 20 imply a decreasing $H_2SO_4$/MSA ratio with rising $HO_2$ concentrations (Fig. 1), which is not visible in the experiments (Supplementary Fig. 3). This means that our experimental findings do not support considerable MSA formation via $CH_3SO_3 + HO_2$ (pathway 20). Moreover, an increasing $H_2SO_4$/MSA ratio with decreasing $CH_3SH$ concentration was observed for otherwise nearly constant reaction conditions, including $CH_3SH$ consumption by the OH reaction and the prevailing $HO_2$ concentration (Fig. 3). The MSA signal practically disappeared for $CH_3SH$ concentrations below a few $10^{10}$ molecules cm⁻³. Thus, the reaction of $CH_3SO_3$ with $CH_3SH$ (pathway 19), likely via H-abstraction of the labile S-bound H atom, seems to dominate the MSA formation under the present conditions. This also means that the direct $H_2SO_4$ formation via $CH_3SO_3$ decomposition (pathway 17) is supressed in

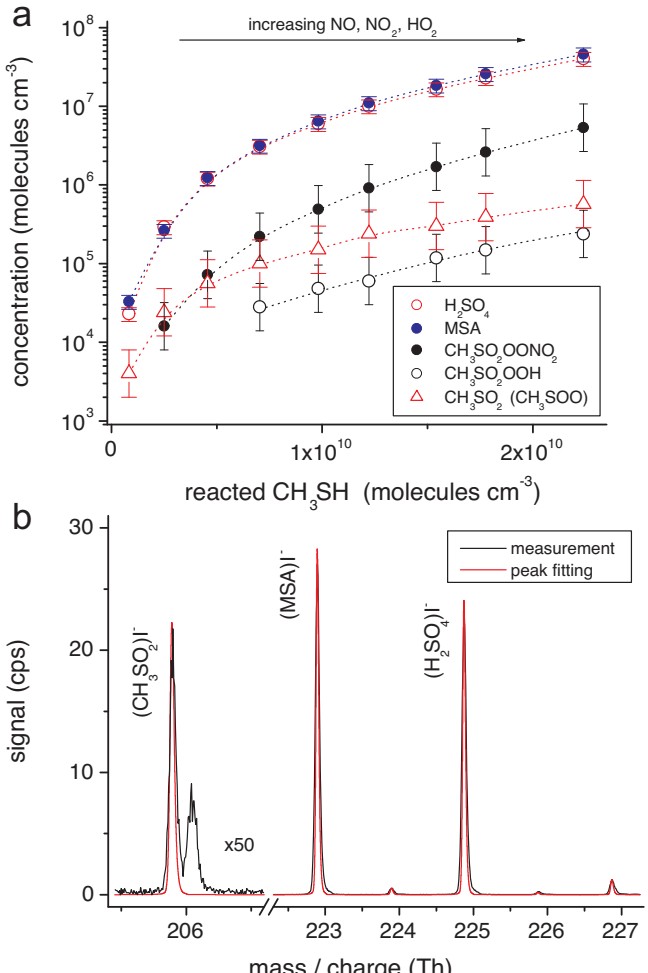

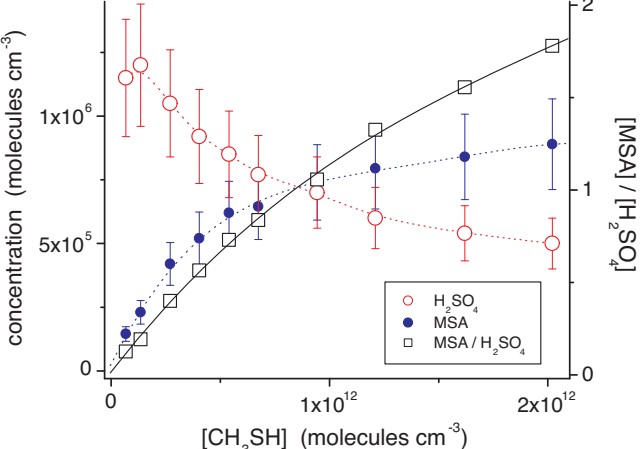

**Fig. 3 | Concentrations of $H_2SO_4$ and MSA (methane sulfonic acid), and the MSA/$H_2SO_4$ ratio as a function of $CH_3SH$ concentration.** Experiments on OH + $CH_3SH$ were carried out in the free-jet flow system at r.h. = 10% and a reaction time of 7.9 s using IPN (isopropyl nitrite) photolysis for OH radical generation. Reactant concentrations are stated in Supplementary Table 1. The error bars for $H_2SO_4$ and MSA depict the uncertainty of ~20% based on the uncertainty in the calibration factor. Source data are provided as a Source Data file.

**Fig. 2 | Product detection from $CH_3S$ oxidation using ionisation by iodide.** Experiments on OH + $CH_3SH$ for $CH_3S$ production have been conducted in the free-jet flow system, t = 7.9 s, at r.h. = 10%. OH radicals were produced from IPN (iso-propyl nitrite) photolysis, i.e. OH radical generation via NO + $HO_2$ → $NO_2$ + OH. Increasing OH radical levels for rising $CH_3SH$ conversion were linked by increasing concentrations of NO, $NO_2$ and $HO_2$ due to increasing IPN conversion in the pho-tolysis. Reactant concentrations are stated in Supplementary Table 1. Source data are provided as a Source Data file. **a** Detected products are given as a function of converted $CH_3SH$. Data for $H_2SO_4$ and MSA (methane sulfonic acid) are based on absolute calibration with an uncertainty of ~20%. Other concentrations represent lower limits with an uncertainty of a factor of 2. **b** Measured raw spectrum from 10 min data accumulation compared with calculated signals of iodide adducts with $CH_3SO_2$, $H_2SO_4$ and MSA from peak fitting.

the presence of sufficiently high $CH_3SH$ or other substances serving as H-atom donor. A similar behaviour of the $H_2SO_4$/MSA ratio was also observed in OH + DMDS experiments varying the DMDS con-centrations (Supplementary Fig. 4). Here, almost exclusive $H_2SO_4$ formation can be expected for DMDS concentrations below $10^{10}$ molecules $cm^{-3}$.

Because atmospheric $CH_3SH$ and DMDS concentrations are clearly smaller than $10^{10}$ molecules $cm^{-3}$ (400 ppt), see attached Sup-plementary Dataset 1 and ref. 5, $CH_3SO_3$ decomposition (pathway 17) forming finally the direct $H_2SO_4$ most likely dominates the fate of $CH_3SO_3$ for atmospheric conditions (Fig. 3 and Supplementary Fig. 4). MSA formation according to $CH_3SO_3$ + RH for RH ≡ $CH_3SH$ or DMDS (pathway 19) has to be of minor importance. It is speculative whether or not other hydrocarbons RH in the atmosphere could efficiently form MSA via pathway 19.

## Formation routes to direct $H_2SO_4$

We evaluated the impact of atmospheric trace gases, i.e. ozone, $RO_2$ radicals, NO and $NO_2$, in the process of direct gas-phase $H_2SO_4$ for-mation with separate experiments starting from the OH radical reac-tions with $CH_3SH$ and DMDS (Fig. 4). While OH + $CH_3SH$ represents a clean source of $CH_3S$ with a reported formation yield of $1.1 ± 0.2$[34], OH + DMDS is expected to form $CH_3S$ and $CH_3SOH$[39,40], which further reacts with ozone leading mainly to $CH_3SO_2$ with a yield close to unity for high enough ozone concentrations[32]. The OH + DMS reaction was not considered in these experiments because of its complexity[9,11], which complicates the investigation of selected pathways. Reaction conditions were chosen in such a way that intermediate concentra-tions were kept as low as possible in order to suppress unwanted bimolecular steps not relevant in the atmosphere. For this reason, the amount of reacted $CH_3SH$ and DMDS was limited to a few $10^8$ mole-cules $cm^{-3}$. Gas-phase $H_2SO_4$ formation starting from the reaction of $SO_2$ with OH radicals or Criegee intermediates was again small in these measurement series and did not influence the results of direct $H_2SO_4$ formation significantly, see also Methods.

Ozone: No significant $H_2SO_4$ formation from OH + $CH_3SH$ was observed for ozone concentrations of up to $2 × 10^{12}$ molecules $cm^{-3}$ (~80 ppb) in the free-jet flow system with the short reaction time of 7.9 s. $H_2SO_4$ became detectable in the laminar flow tube (LFT) with a reaction time of 32 s indicating a relatively slow process of direct $H_2SO_4$ formation (Fig. 4a). Big differences in the $H_2SO_4$ yields of more than an order of magnitude were measured using either OH + $CH_3SH$ for $CH_3S$ generation or OH + DMDS forming $CH_3S$ and most likely $CH_3SO_2$ with high yields. Considering $CH_3SO_2$ as the needed inter-mediate for direct $H_2SO_4$ formation (Fig. 1), $CH_3S's$ oxidation obviously proceeds only with a small share via $CH_3SO_2$, e.g. ≤ 9% for an ozone concentration of $5.7 × 10^{11}$ molecules $cm^{-3}$ (Fig. 4a) taking OH + DMDS with a $CH_3SO_2$ yield of unity as the reference. Moreover, OH + $CH_3SH$ experiments with heavy ozone ($^{18}O_3$) revealed the absence of $H_2SO_4$ containing three $^{18}O$ atoms (Supplementary Fig. 5) as expected from the reaction sequence 3, 6b and 9 (Fig. 1). We largely measured $H_2SO_4$ with one $^{18}O$ atom consistent with the reaction sequence 1/−1, 2 and 9. The findings imply the dominance of pathway 6a over 6b or in fact the irrelevance of pathway 6b, allowing for the importance of ozone reactions in the $CH_3S$ oxidation[8,34,41]. This can be

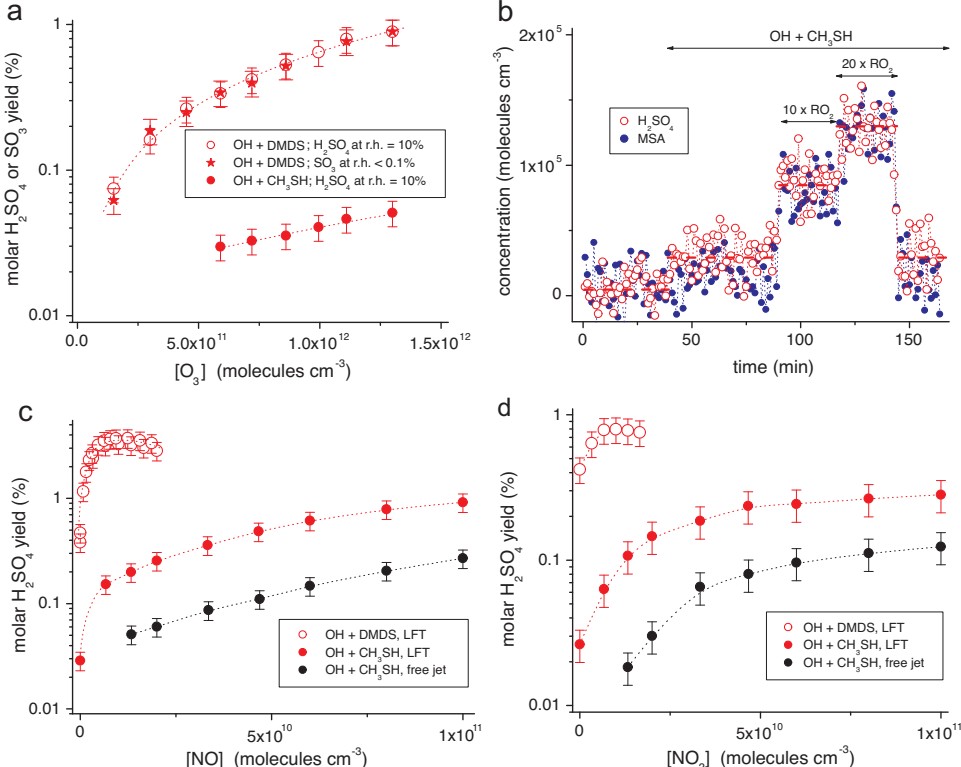

**Fig. 4 | Impact of trace gases on direct $H_2SO_4$ formation using ionisation by nitrate.** Experiments were conducted either in the free-jet flow system, t = 7.9 s, or in the laminar flow tube (LFT), t = 32 s, at r.h. = 10% (or <0.1%) using tetramethyl ethylene (TME) ozonolysis for OH radical production. Reactant concentrations are given in Supplementary Table 1. Error bars represent the uncertainty of ~20% in the absolute calibration. Source data are provided as a Source Data file. **a** Formation of $H_2SO_4$ and $SO_3$ as a function of ozone measured in the LFT. Reacted $CH_3SH$ was in the range $(7.6–17) \times 10^7$ and $(2.1–18) \times 10^7$ molecules cm$^{-3}$ for DMDS (dimethyl

disulfide). **b** $H_2SO_4$ and MSA (methane sulfonic acid) formation from OH + $CH_3SH$ depending on $RO_2$ radical concentrations, $CH_3C(O)CH_2O_2$ and $CH_3O_2$, in the LFT. Rising $RO_2$ levels were achieved by stepwise increase of TME and corresponding $CH_4$ additions keeping reacted $CH_3SH$ constant at ~$7.6 \times 10^7$ molecules cm$^{-3}$. Highest $CH_3C(O)CH_2O_2$ and $CH_3O_2$ concentrations were $1.0 \times 10^9$ and $8.8 \times 10^8$ molecules cm$^{-3}$, respectively, calculated from an extended model (Supplementary Table 4). **c** $H_2SO_4$ formation yields as a function of NO addition. **d** $H_2SO_4$ formation yields as a function of $NO_2$ addition.

explained by the high exothermicity of the $CH_3SO + O_3$ reaction forming the chemically excited $CH_3SO_2^*$ that rapidly decomposes to $SO_2$ and $CH_3$ before it is thermalised[42].

The small $H_2SO_4$ yields < 1% for atmospheric ozone concentrations, even under conditions of the preferred $CH_3SO_2$ generation from OH + DMDS, support the efficient decomposition of $CH_3SO_2$ (pathway 8), which is in line with the high $SO_2$ yields reported recently[32,33]. $SO_3$ yields measured under dry conditions, r.h. < 0.1%, were in very good agreement with the $H_2SO_4$ yields at r.h. = 10% (Fig. 4a) confirming $SO_3$ as the precursor of $H_2SO_4$ (pathways 17 and 18 in Fig. 1).

An ozone concentration of $5.7 \times 10^{11}$ molecules cm$^{-3}$ (~23 ppb) was chosen in the following experiments, which stands for an average concentration over pristine oceans[43], making our findings applicable to the atmospheric reaction system.

$RO_2$ radicals: We detected a distinct impact of $RO_2$ radicals on the formation of $H_2SO_4$ and MSA (Fig. 4b). Main $RO_2$ radicals in the reaction system are $CH_3C(O)CH_2O_2$, formed in the course of OH radical generation via TME ozonolysis[44,45], and $CH_3O_2$ as the by-product of $SO_2$ in the oxidation of $CH_3SH$ and DMDS[32,33] as well as from OH + $CH_4$ in the case of $CH_4$ additions. In the OH + $CH_3SH$ reaction, we increased in a two-step process the concentrations of $CH_3C(O)CH_2O_2$ and $CH_3O_2$ radicals, first by a factor of ~10, i.e. from $6.2 \times 10^7$ to $5.7 \times 10^8$ and from $4.0 \times 10^7$ to $4.4 \times 10^8$ molecules cm$^{-3}$, respectively, leading to enhanced $H_2SO_4$ formation by a factor of ~3.5 for constant $CH_3SH$ conversion (Fig. 4b). Further doubling of the $RO_2$ concentrations led to further rise in $H_2SO_4$ productions. The MSA formation, however, increased stronger than that of $H_2SO_4$, which became more visible from a similar experiment on OH + DMDS (Supplementary Fig. 6). Furthermore, we

observed predominate MSA formation in a reaction system with HO-$C_6H_{12}O_2$ along with $CH_3C(O)CH_2O_2$ as the main $RO_2$ radicals (Supplementary Fig. 7). It can be speculated that most likely $CH_3SO_2OO$ reacted with $RO_2$ radicals either via the alkoxy channel (pathway 15 in Fig. 1), forming finally $H_2SO_4$, or via the dismutation channel (pathway 16 in Fig. 1), similar to the well-known chemistry of carbon-centred $RO_2$ radicals[46], leading to MSA. The branching ratio of pathways 15 vs. 16 appears to be dependent on the structure of the reacting $RO_2$ radical. Other $RO_2$ driven pathways, influencing the product formation, cannot be ruled out.

Nitrogen oxide (NO): Addition of NO substantially accelerated the $H_2SO_4$ formation in all experiments (Fig. 4c) supporting the potential importance of $CH_3SO_2OO$ for $H_2SO_4$ formation, here via pathway 14 (Fig. 1). An increase in the $H_2SO_4$ production by a factor of ~4 (Supplementary Fig. 8) was measured using a NO concentration of $1 \times 10^9$ molecules cm$^{-3}$ similar to the behaviour observed for elevated $RO_2$ concentrations (Fig. 4b and Supplementary Fig. 6). This indicates rate coefficients $k_{14}$ and $k_{15}$ for the reaction of $CH_3SO_2OO$ with NO and $RO_2$, respectively, being in the same range. Comparison of results for relatively low NO concentrations of < $10^{10}$ molecules cm$^{-3}$ in the LFT showed more than one order of magnitude higher $H_2SO_4$ yields from the oxidation of DMDS relative to $CH_3SH$, in line with the findings from the pure ozone-driven reaction (Fig. 4a). For elevated NO levels, other NO reactions presumably disturbed the $CH_3SO_2$ formation from OH + DMDS and inhibited further rise of the $H_2SO_4$ yield. Further NO reactions in the $CH_3S$ oxidation could also negatively impact the $H_2SO_4$ formation, such as $CH_3S + NO \rightarrow CH_3SNO$[10] or $CH_3SOO + NO \rightarrow CH_3SO + NO_2$[10] (pathway 5) resulting finally in $SO_2$ formation via

pathway 6a (Fig. 1). The higher $H_2SO_4$ yields from $OH + CH_3SH$ measured in the LFT at t = 32 s point again to a slow process of $H_2SO_4$ formation that is far away from completeness for the reaction time of 7.9 s in the free-jet flow system.

Nitrogen dioxide ($NO_2$): Addition of $NO_2$ featured a similar effect for the rise of $H_2SO_4$ yields (Fig. 4d) as observed for NO (Fig. 4c), albeit the $NO_2$ impact was less pronounced. It is supposed in the literature that $NO_2$ reacts with $CH_3SO_2$ forming $CH_3SO_3$ (pathway 10) which finally leads to $H_2SO_4$ analogous to the ozone-mediated route (pathway 9)[9]. This set of experiments confirmed again the much higher potential of $H_2SO_4$ formation starting from $OH + DMDS$ regarding $OH + CH_3SH$, as well as the slow formation rate of the direct $H_2SO_4$ production. $H_2SO_4$ production almost doubled as the result of a $NO_2$ addition of $6.7 \times 10^9$ molecules $cm^{-3}$ in the LFT experiments (Fig. 4d), indicating nearly the same reaction rate in the reaction of $CH_3SO_2$ with ozone and $NO_2$ (pathways 9 and 10 in Fig. 1), $[O_3] = 5.7 \times 10^{11}$ molecules $cm^{-3}$. This leads to $k_9/k_{10} \sim 1/85$ being in good agreement with the rate coefficient ratio currently used in models[21,22]. The experiments with $NO_2$ addition did not allow any conclusions regarding the relative importance of the product channels 7a and 7b from $CH_3SO + NO_2$.

In summary, the experiments provided evidence for the promoting effect of each of the four important trace gases for the direct $H_2SO_4$ formation. The relatively strong impact caused by $RO_2$ radicals and NO (Fig. 4b and 4c) was surprising, which further highlights $CH_3SO_2OO$ radicals as important intermediates.

## Application to the atmosphere

Adjustments in the $H_2SO_4$ yields were needed in order to apply the laboratory findings for atmospheric conditions. The incompleteness of the $CH_3SO_3$ conversion due to the short reaction times led to a correcting factor of 1.6 for the $H_2SO_4$ yields in the LFT using $k_{17} = 0.076\ s^{-1}$ at $295 \pm 2\ K$, see Methods. Relatively high $CH_3SH$ and DMDS concentrations in the experiments, not present in the atmosphere, necessitated further adjustment by a factor of $\sim 1.5$ to allow for the suppression of $H_2SO_4$ formation in their competing reaction with $CH_3SO_3$ forming MSA (pathway 19 in Fig. 1), see Fig. 3 and Supplementary Fig. 4. Adjusted $H_2SO_4$ yields for low-$NO_x$ conditions and $[O_3] = 5.7 \times 10^{11}$ molecules $cm^{-3}$ were estimated to be $0.074 \pm 0.015\%$ per formed $CH_3S$ and $0.82 \pm 0.02\%$ per formed $CH_3SO_2$ (Fig. 4a) assuming a $CH_3SO_2$ yield of unity from $OH + DMDS$. The yields increased to $0.11 \pm 0.02\%$ ($CH_3S$) and $1.2 \pm 0.2\%$ ($CH_3SO_2$) incorporating the "$RO_2$ effect" (Fig. 4b) for total $RO_2$ radical concentrations of $\sim 3 \times 10^8$ molecules $cm^{-3}$, that represents an average $RO_2$ level during main $CH_3S$ and $CH_3SO_2$ production at noon (Supplementary Fig. 9). The ratio $k_8/(k_9 \times [O_3]) \sim 120$ ($k_8/k_9 \sim 7 \times 10^{13}$ molecules $cm^{-3}$) followed from the ozone-driven experiments on $OH + DMDS$ with a $H_2SO_4$ yield of 0.82%, that strongly favours $SO_2$ formation from $CH_3SO_2$ (pathway 8) being consistent with the high $SO_2$ yields measured[32,33]. This $k_8/k_9$ ratio, however, is in contrast to the implementation in latest atmospheric models, $k_8/k_9 = 9.5 \times 10^{11}$ molecules $cm^{-3}$ [21,22], leading to severe overestimation of the modelled $CH_3SO_3$ production.

## Atmospheric impact

Process model simulations were performed with a complex multiphase chemistry mechanism MCM/CAPRAM[47,48] for six different scenarios (Methods and Supplementary Table 2) to assess the importance of the direct gas-phase formation pathway of $H_2SO_4$ from DMS and $CH_3SH$ oxidation relative to the $OH + SO_2$ reaction under pristine marine conditions. Oxidation of DMDS was neglected because of its relatively small emission[5]. The model is able to simulate typical DMS and $SO_2$ mixing ratios (see Supplementary Figs. 10 and 11) as measured under marine conditions (Fig. 5a and attached Supplementary Dataset 1) independent of $NO_x$ levels assumed in the simulations (Supplementary Fig. 12). The modelled $CH_3S$ and $CH_3SO_2$ formation rates are provided in Supplementary Table 3 for all six

simulations. It can be seen that $NO_x$ can likely affect the $CH_3S$ formation, but it is less important for $CH_3SO_2$ formation. The strongest impact on $CH_3SO_2$ production has the $H_A$ applied. The modelled $CH_3S$ and $CH_3SO_2$ formation rates together with the experimental $H_2SO_4$ yields of 0.11% ($CH_3S$), 1.2% ($CH_3SO_2$) and 100% ($SO_2$) were used to calculate the gas-phase $H_2SO_4$ formation rates from the different oxidation pathways and their relative contributions (Fig. 5b). For more clarity, the simulations with higher $NO_x$ are not depicted, because of the modelled low effect on $CH_3S$ and $CH_3SO_2$ formation rates in comparison with the simulation using the smaller NO emission. The data in Supplementary Table 3 reveal that the modelled $CH_3S$ and $CH_3SO_2$ formation rates are only weakly affected by ten times higher $NO_x$ emissions, whereas the considered uptake parameters are the most important influencing factors. As the result, the direct gas-phase formation of $H_2SO_4$ arises mainly from the DMS addition channel and can contribute up to $\sim 50\%$ to the overall gas-phase $H_2SO_4$ production. This emphasises the importance of the direct gas-phase formation route for marine conditions. Fully neglecting the share from the DMS addition channel because of inconsistent $CH_3SO_2$ yields currently in the literature[9,19,49], still a fraction of up to $\sim 12\%$ remains (Fig. 5b). It should be noted, that total direct gas-phase $H_2SO_4$ formation rates exclusively simulated by the model (Supplementary Fig. 13) exceeded those from the combined experiment/model approach (Fig. 5b) by about two orders of magnitude. A main reason for that is the inappropriate description of $CH_3SO_2$´s fate in the latest models[21,22].

The simulations indicate that the concentration ratio of $SO_2$ relative to the reduced-sulfur compounds, mainly DMS, is a critical parameter for the importance of the direct gas-phase $H_2SO_4$ formation. This becomes apparent using data from a field campaign at Baring Head, New Zealand[50], with an air mass change from anthropogenically influenced, $SO_2/DMS > 10$, to the clean pristine ocean regime, $SO_2/DMS < 0.1$ (Fig. 5c). Significant relevance of the direct gas-phase $H_2SO_4$ route only exists for $SO_2/DMS \leq 0.3$. Thus, the direct gas-phase route could be important especially in the Southern Hemisphere, due to low $SO_2/DMS$ ratios existing there (Fig. 5a), and in the outflow of convective marine clouds where $SO_2$ is reduced by scavenging and cloud chemistry.

Field studies often indicated new particle formation in the direct vicinity above marine clouds[51,52] most likely connected to the high OH radical[53] and $H_2SO_4$ concentrations[54] observed at such locations. Updrafts of clouds can inject DMS into the free troposphere. There, we suggest that the direct gas-phase $H_2SO_4$ formation DMS → $H_2SO_4$, as identified in the present study, play an important role for gas-phase $H_2SO_4$ production in cloud outflows, because of the expected low $SO_2/DMS$ ratio immediately after cloud passage and the slow overall process DMS → $SO_2$ → $H_2SO_4$. This implies that directly formed gas-phase $H_2SO_4$ from DMS oxidation is likely substantial for the observed new particle formation in cloud outflows, thereby affecting or even controlling the amounts of CCN available[51,52,55]. Therefore, this study provides the impetus for further developments to incorporate and study such processes in regional and global atmospheric chemistry transport as well as climate models.

In conclusion, we experimentally demonstrated the direct formation of $H_2SO_4$ in the course of atmospheric gas-phase oxidation of reduced-sulfur compounds. We found strong indications for the reactions $CH_3SO_2 + O_3$ (pathway 9) and $CH_3SO_2OO + RO_2$ (pathway 15) being the rate limiting steps for $H_2SO_4$ production under low-$NO_x$ conditions. The strong increase of $H_2SO_4$ production in the presence of NO emphasises the role of $CH_3SO_2OO$ radicals in this reaction system. Our findings do not support considerable MSA formation via the $CH_3SO_3 + HO_2$ pathway.

Although the direct $H_2SO_4$ formation yields appear to be pretty small, for concentration ratios $SO_2/DMS \leq 0.3$, i.e. for conditions as encountered especially over the oceans in the Southern

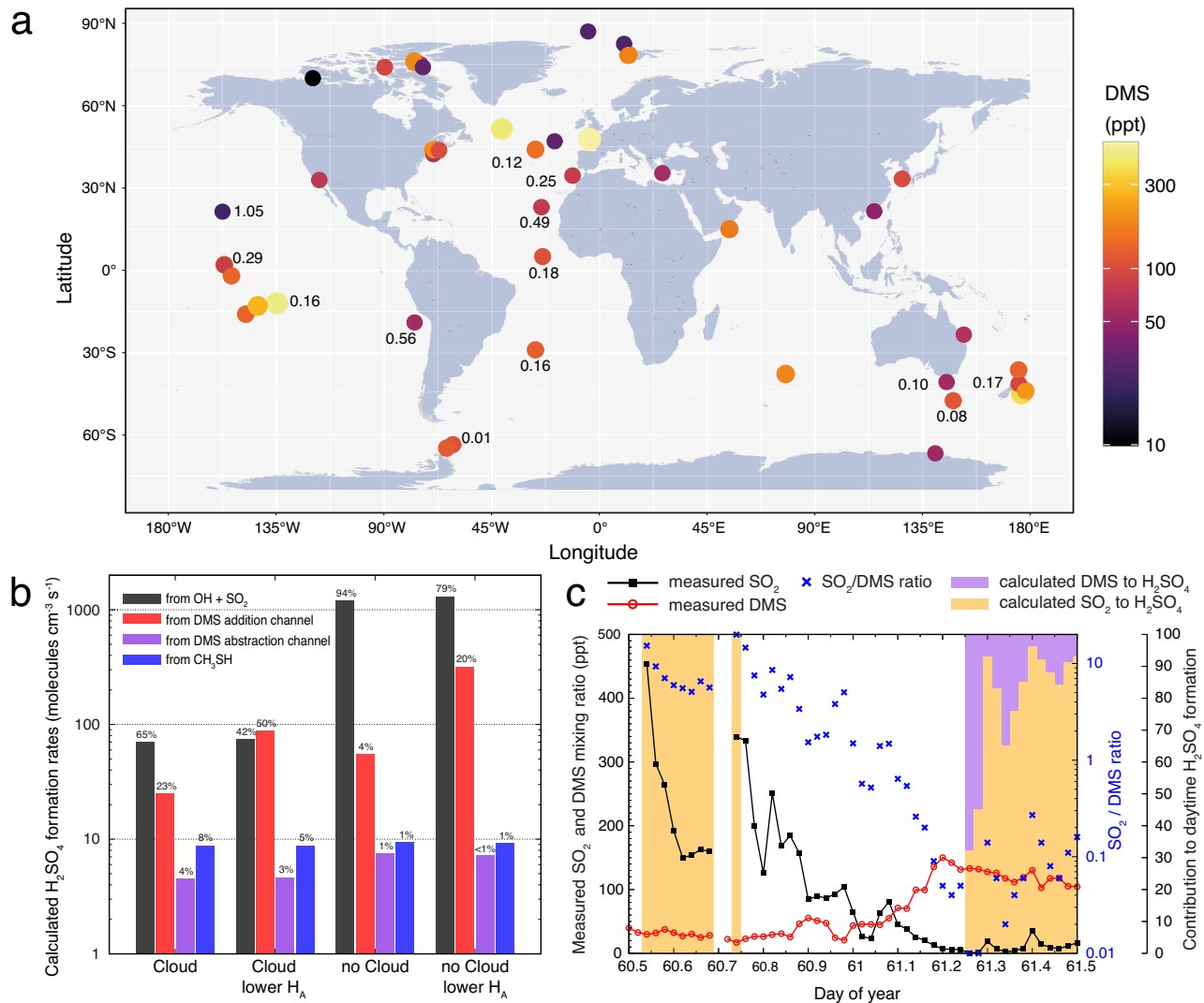

**Fig. 5 | Measured DMS (dimethyl sulfide) and SO₂ data and the contribution of different pathways to H₂SO₄ formation.** Source data are provided as a Source Data file. **a** Measured average DMS concentrations (colour coded) worldwide with corresponding SO₂/DMS ratios (numbers), if available. The map plot was created with R[68] using the ggplot2 package (map_data("world")). **b** Calculated H₂SO₄ formation rates from different pathways combining the modelled CH₃S and CH₃SO₂ formation rates with experimentally determined H₂SO₄ yields. The "Cloud" scenario represents simulations with cloud passages leading to lower SO₂ concentrations due to its uptake and oxidation in clouds. The simulations with "lower H$_A$" represent model runs using lower Henry constants H$_A$ of DMS oxidation products (see Supplementary Table 1). **c** Reproduction of measured SO₂ (black line with squares) and DMS (red line with dots) concentrations as well as its SO₂/DMS ratio (blues crosses) from observations at Baring Head, New Zealand[50]. Purple and yellow bars illustrate the calculated relative contributions of DMS (direct route) and SO₂ oxidation to total H₂SO₄ formation at daytime. The contributions are calculated based on results from "no Cloud" simulation.

Hemisphere, and/or in the outflows of clouds, the direct route could be competitive with the established OH + SO₂ path of H₂SO₄ generation.

All in all, we herewith suggest direct gas-phase formation of H₂SO₄ from reduced-sulfur compounds, such as DMS, to be an atmospherically relevant process for the production of H₂SO₄, and consequently for the formation of new particles, under, e.g. the pristine marine conditions in the Southern Hemisphere.

## Methods

### Experimental setup

The investigations were carried out in two flow systems, i.e. in the free-jet flow system[30,31] and the laminar flow tube (LFT)[29] at 1 bar of air and a temperature of 295 ± 2 K. The flow tubes worked with different residence times, 7.9 and 32 s, respectively, that allowed to draw a conclusion regarding the rate of relatively slow processes, here on the thermal decomposition of CH₃SO₃.

The free-jet flow system consists of an outer tube (length: 200 cm, inner diameter: 16 cm) and a moveable inner tube (outer diameter: 9.5 mm) connected with a nozzle of 3 mm inner diameter. The first reactant (isopropyl nitrite (IPN) or ozone) premixed with air (5 litre min⁻¹, STP) was injected through the inner tube into the main gas stream (95 litre min⁻¹, STP), which contained the second reactant (CH₃SH and/or tetramethyl ethylene (TME) along with additions if needed) diluted in air. Downstream the nozzle, large differences of the gas velocities at the nozzle outflow (nozzle: 15.9 m s⁻¹; main flow: 0.13 m s⁻¹) and the nozzle geometry ensured rapid turbulent reactant mixing[56]. The reaction time was 7.9 s, experimentally obtained by a "chemical clock". This set-up allows to carry out investigations under atmospheric conditions in absence of wall effects. IPN photolysis[57] for continuous OH radical generation in the flow system was conducted downstream the mixing point of the reactants by means of 8 NARVA 36 W Blacklight Blue lamps emitting in the range 350–400 nm. The photolysis of IPN produces NO and i-C₃H₇O radicals, which rapidly

formed acetone and $HO_2$ radicals in the reaction with $O_2$. OH radical generation finally took place via $HO_2 + NO \rightarrow OH + NO_2$. Ozonolysis of TME[44] served as non-photolytic OH radical source under low-$NO_x$ conditions.

The laminar flow tube (LFT) (i.d. 8 cm; total length 425 cm) consists of a first section (56 cm) containing the gas inlet system, a second middle section (344 cm) representing the reaction zone and an end section (25 cm) incorporating the sampling devices. Here, TME ozonolysis was exclusively used for OH radical formation. Ozone was injected through a nozzle system into the gas mixture, containing TME, reduced-sulfur compounds and additives if needed, just before entering the middle section. The total flow was set at 30 litre min$^{-1}$ (STP) resulting in a residence time of 32 s in the reaction zone.

Humidified air in both setups was supplied by flushing a part of the air flow through three water saturators filled with water from an ultrapure water system (Barnstead, resistivity: 17.4 MΩ cm). The relative humidity of the reaction gas was continuously controlled at the outflow by a humidity sensor (Hygrosense HYTE). Ozone was monitored by a gas monitor (Thermo Scientific iQ 49) and the concentration of organic compounds by a proton transfer reaction - mass spectrometer (Ionicon, high sensitivity PTR-MS)[58]. The "crude" air was taken from a pressure swing adsorption unit with further purification by means of absorber units filled with charcoal, a hopcalite (CuMnO$_x$) catalyst[59] and different activated 4 Å and 10 Å molecular sieves.

## Reactant concentrations and conversion and the importance of OH + SO$_2$ and Criegee intermediate + SO$_2$

Initial reactant concentrations are either given in Supplementary Table 1 for the experiments described in the main text, Figs. 2–4, or in the figure captions. The amount of reacted $CH_3SH$ in the IPN photolysis experiment (Fig. 2a) was measured in an additional run in the presence of $SO_2$ (for otherwise identical reaction conditions) by monitoring $H_2SO_4$ formation. The $SO_2$ concentration, $7.5 \times 10^{11}$ molecules cm$^{-3}$, was chosen to such an extent that only 2% of formed OH radicals reacted with $SO_2$ and, thus, the product formation of the OH + $CH_3SH$ reaction was not disturbed[12]. Reacted $CH_3SH$ is available from the measured $H_2SO_4$ (after correction of the fraction arising from $CH_3SH$ oxidation) considering the OH reactivity in the parallel reactions OH + $CH_3SH$ and OH + $SO_2$. In the case of TME ozonolysis for OH production, the amount of reacted $CH_3SH$ or DMDS was calculated based on a detailed reaction scheme (Supplementary Table 4). Modelling calculations, including the IPN photolysis experiment, confirmed that $H_2SO_4$ production starting from the reaction of $SO_2$ with OH radicals or Criegee intermediates did not significantly influence the results of direct $H_2SO_4$ formation from the organic sulfur compounds.

## Mass spectrometric analysis

Detection of $H_2SO_4$, $CH_3SO_3H$ and other oxidation products was carried out using a CI-APi-TOF (chemical ionisation - atmospheric pressure interface - time-of-flight) mass spectrometer with a resolving power > 3000 Th/Th (Tofwerk) connected to a Boulder-type inlet system (Airmodus) operating with iodide (I$^-$) and nitrate (NO$_3^-$) as the reagent ions at atmospheric pressure[12,31].

In the case of ionisation by iodide, tert-butyl iodide premixed in a flask was added to a 35 litre min$^{-1}$ (STP) sheath flow of purified nitrogen leading to a tert-butyl iodide concentration of $4.8 \times 10^{11}$ molecules cm$^{-3}$. Produced ions after ionisation with a $^{241}$Am source were I$^-$ and to a lesser amount I(H$_2$O)$^-$. The ions from the sheath flow were guided into the sample flow by an electric field without mixing of both gas streams. In the case of ionisation by nitrate, an HNO$_3$ containing vial was connected to the 35 litre min$^{-1}$ (STP) flow without overflowing the HNO$_3$ sample. HNO$_3$ diffusion from the vial was found to be sufficient to form the reagent ions (HNO$_3$)$_x$NO$_3^-$, x = 0, 1, 2, after ionisation.

Absolute signal calibration was used in the measurements of $H_2SO_4$ applying iodide and nitrate ionisation as well as in the

determination of $SO_3$, which was detected as the adduct ($SO_3$)NO$_3^-$ and $SO_4^-$ [60,61] using nitrate ionisation. $H_2SO_4$ and $SO_3$ production in the calibration experiments for wet (r.h. = 10%) and dry (r.h. <0.1%) conditions, respectively, was carried out via TME ozonolysis in the presence of $SO_2$[62]. The calibration factors obtained for $H_2SO_4$ were also taken for $CH_3SO_3H$. In the case of $CH_3SO_2$, $CH_3SO_2OOH$ and $CH_3SO_2OONO_2$, a calculated calibration factor of $2 \times 10^9$ molecules cm$^{-3}$ was taken, resulting in lower limit concentrations for these products with an uncertainty of a factor of two[31,45].

## Kinetic data analysis

$H_2SO_4$ and MSA wall loss in the LFT:
The rate law for $H_2SO_4$ is given by

$$\frac{d[H_2SO_4]}{dt} = P_{H_2SO_4} - k_{loss} \times [H_2SO_4] \tag{1}$$

assuming a time-independent production term of $H_2SO_4$, $P_{H_2SO_4}$. This assumption is justified because of constant OH radical production during the whole reaction time and practically constant reactant concentrations due to reactant conversions clearly smaller than 1% in each case. Integration of Eq. (1) with $[H_2SO_4]_{t=0} = 0$ yields:

$$[H_2SO_4]_t = \frac{P_{H_2SO_4}}{k_{loss}} \left(1 - \exp(-k_{loss} \times t)\right) \tag{2}$$

$[H_2SO_4]_t = P_{H_2SO_4} \times t$ follows for the wall-loss free $H_2SO_4$ concentration. Consequently, the relative $H_2SO_4$ loss in the tube is given by:

$$H_2SO_4\,loss = 1 - \frac{1}{k_{loss} \times t} \left(1 - \exp(-k_{loss} \times t)\right) \tag{3}$$

The value of $k_{loss}$ can be described by the diffusion-controlled wall-loss term $\frac{3.65 \times D}{r^2}$ using an experimentally obtained $H_2SO_4$ diffusion coefficient D = 0.08 cm$^2$ s$^{-1}$ [63] leading to $k_{loss} = 0.018$ s$^{-1}$. Based on that and for the reaction time of 32 s in the LFT, a $H_2SO_4$ loss of 24% was calculated using Eq. (3). Thus, the measured $H_2SO_4$ concentration was multiplied with 1.315 to consider the wall loss. The same was applied for MSA.

Determination of $k_{17}$ describing $CH_3SO_3 \rightarrow SO_3 + CH_3$:
Assuming dominant loss of $CH_3SO_3$ via its decomposition into $SO_3$ and $CH_3$ and a time-independent production term of $CH_3SO_3$, $P_{CH_3SO_3}$, due to practically constant reactant conditions, the rate law of $CH_3SO_3$ is given by:

$$\frac{d[CH_3SO_3]}{dt} = P_{CH_3SO_3} - k_{17} \times [CH_3SO_3] \tag{4}$$

Integration of Eq. (4) with $[CH_3SO_3]_{t=0} = 0$ yields:

$$[CH_3SO_3]_t = \frac{P_{CH_3SO_3}}{k_{17}} \left(1 - \exp(-k_{17} \times t)\right) \tag{5}$$

The rate law of $SO_3$ formation is

$$\begin{aligned}\frac{d[SO_3]}{dt} &= k_{17} \times [CH_3SO_3] \\ &= P_{CH_3SO_3} \times (1 - \exp(-k_{17} \times t))\end{aligned} \tag{6}$$

leading after integration with $[SO_3]_{t=0} = 0$ to:

$$\frac{[SO_3]_t}{P_{CH_3SO_3}} = t - \frac{1}{k_{17}} \left(1 - \exp(-k_{17} \times t)\right) \tag{7}$$

 7

Because of immediate $SO_3$ conversion to $H_2SO_4$ under humid conditions, Eq. (7) can be written in the following way:

$$\frac{[H_2SO_4]_t}{P_{CH_3SO_3}} = t - \frac{1}{k_{17}}\left(1 - \exp(-k_{17} \times t)\right) \qquad (8)$$

Equation (8) was used to determine $k_{17}$ based on measured $H_2SO_4$ concentrations from $OH + CH_3SH$ depending on $NO_2$ additions in both flow systems, i.e. in the free-jet flow system with t = 7.9 s and in the LFT with t = 32 s for otherwise similar conditions (Supplementary Fig. 14 and Fig. 4d). The ratio $[H_2SO_4]_{32s}/[H_2SO_4]_{7.9s}$ was found to be 4.5 ± 0.6 with a corresponding ratio of the $CH_3SO_3$ production rates $P_{CH_3SO_3}$ of 1/2.27, that considered the different reactant concentrations in the experiments and the fraction of OH radicals reacting with $CH_3SH$. For the mean $H_2SO_4$ ratio of 4.5, $k_{17} = 0.076\,s^{-1}$ was calculated leading to $0.076^{+0.034}_{-0.025}\,s^{-1}$ that involves the bounds of the $H_2SO_4$ ratio. Based on Eq. (9), which corresponds to Eq. (3) for $H_2SO_4$ loss,

$$CH_3SO_3\ decomposition = 1 - \frac{1}{k_{17} \times t}\left(1 - \exp(-k_{17} \times t)\right) \qquad (9)$$

it was possible to calculate with $k_{17} = 0.076\,s^{-1}$, that 62% of formed $CH_3SO_3$ decomposed in the LFT (t = 32 s) making a correction factor of 1.6 necessary in order to account for total removal via pathway 17. It is to be noted, that the reaction of $CH_3SH$ with $CH_3SO_3$, pathway 19, does not influence this result as long as the contribution of this pathway (same $CH_3SH$ concentration) is identical in both flow experiments.

## Atmospheric modelling

Complex multiphase chemistry simulations were performed using the SPectral Aerosol Cloud Chemistry Interaction Model (SPACCIM)[64] to study the contributions of different reaction pathways from DMS, $SO_2$ and $CH_3SH$ leading to sulfuric acid or its precursors under pristine marine conditions. It should be noted, that the applied model is not designed to simulate new particle formation. Thus, nucleation driven by gas-phase $H_2SO_4$ and resulting effects cannot be investigated. Therefore, we only focus on the chemical gas-phase $H_2SO_4$ formation in the present study.

In the model, the multiphase chemistry is described by combining the near-explicit gas-phase mechanism MCMv3.2[65,66] and detailed aqueous-phase chemistry mechanism CAPRAM4.0 (Chemical Aqueous Phase RAdical Mechanism version 4.0)[47], respectively. This mechanism system describes the formation of gas-phase $H_2SO_4$ and aqueous sulfate in a very detailed manner. The representation of the specific multiphase chemistry of reactive halogen species and dimethyl sulfide, important for the marine atmosphere, is achieved through two additional reaction modules, CAPRAM–HM3.0[48] and CAPRAM–DM1.0[21]. With these two additional modules, CAPRAM4.0 includes almost all known sulfate formation pathways in the atmospheric aqueous phase, such as S(IV) oxidation by $H_2O_2$, $O_3$, $HNO_4$, reactive halogen species ($X_2^-$ radical or HOX, with X = Cl, Br or I) or transition metal ions.

Due to the intended foci of the simulations, the complex multiphase DMS chemistry scheme of CAPRAM-DM1.0 has been upgraded with recent mechanism updates and an oxidation scheme for $CH_3SH$ was implemented (Supplementary Table 5). The mechanistic updates comprise the formation of the hydroperoxymethyl thioformate (HPMTF) and its further oxidation in the gas phase. The gas-phase HPMTF oxidation follows mainly the proposed routes described by Wu et al. (2015)[11], considering $SO_2$ or OCS formation, only. Thus, HPMTF cannot contribute to the direct gas-phase formation of $H_2SO_4$ in this mechanism. Phase transfer and subsequent aqueous-phase processing of HPMTF, not included yet because of the current high uncertainties, do not change this. Briefly, the revised mechanism scheme contains 128 gas-phase reactions, 5 phase transfer processes and 50 aqueous-phase reactions.

In the process simulations, an aerosol particle spectrum representative for pristine marine conditions is included[67]. The whole model setup (emission, deposition, initialisation of the gas-phase and particle-phase composition) is the same as applied in previous DMS chemistry studies[21,48]. An exception is the newly included emission of $CH_3SH$ (emission rate: $3.18 \times 10^3$ molecules $cm^{-3}\,s^{-1}$), which is a factor of ten lower compared to that of DMS. This difference is in line with field measurements, see attached Supplementary Dataset 1.

In total, six simulations were performed, separated into (i) three with ("Cloud") and (ii) three without ("no Cloud") cloud processing along the air parcel trajectory. The total simulation time is 108-hours but only day 2 to 4 were considered for data analysis to avoid spin-up effects. Runs are performed for summer conditions, with a boundary layer temperature and relative humidity of 280 K and 70% during non-cloud periods. In the simulations with cloud interactions, eight non-permanent clouds are considered. Every cloud exists for about two hours and occurs around noon and midnight, respectively. Cloud formation is achieved through adiabatic cooling of the air parcel 15 minutes before 11 a.m. and p.m., and the cloud evaporation is realised by adiabatic warming 15 minutes after 1 p.m. and a.m., respectively. Besides the two microphysical scenarios ("Cloud", "no Cloud"), simulations were run with Henry's Law constants implemented in the base mechanism[21] and with lower Henry's Law constants for DMSO ($H_{A,298K} = 2.43 \times 10^5$ mol atm$^{-1}$), DMSO$_2$ ($H_{A,298K} = 1.18 \times 10^6$ mol atm$^{-1}$), and MSIA ($H_{A,298K} = 1.69 \times 10^6$ mol atm$^{-1}$) calculated by COSMOtherm[22]. The two uptake cases were performed to consider the potential uncertainty in the Henry's Law constants and to study their impact of the sulfuric acid formation. In addition, simulations with an increased NO emission by a factor of ten were run using Henry's Law constants implemented in the base mechanism. All simulation scenarios together with their individual configurations are outlined in Supplementary Table 2.

For the six different simulations, averaged net rates (in molecules $cm^{-3}\,s^{-1}$) for the daytime oxidation of DMS, $CH_3SH$ and $SO_2$ between the second to the fourth model day were calculated as well as primary daytime production rates of $CH_3S$ and $CH_3SO_2$. For the oxidation of DMS, we distinguish between rates of the addition and abstraction pathways. All calculated rates are given in Supplementary Table 3.

## Data availability

The measurement data collected from the literature and used in this work are provided in the attached Supplementary Dataset 1. The data generated in this study are provided in the Supplementary Information. Source data are provided with this paper.

## Code availability

The code of MCM is provided via http://mcm.leeds.ac.uk/MCM and CAPRAM code is available at https://capram.tropos.de/. The new and updated mechanism implementation data are provided in the Supplementary Information.

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

## Acknowledgements

The authors thanks K. Pielok, R. Gräfe and A. Rohmer for technical assistance and the tofTools team for providing the data analysis tools. T.B. thanks J. Chen and H.G. Kjaergaard, University of Copenhagen, for motivating discussions. Support came from the Deutsche Forschungsgemeinschaft, project ADOniS (grant HE 3086/53-1, T.B., E.H.H., A.T. and H.H.).

## Author contributions

T.B. designed and carried out the experiments and did the data analysis. F.S. and T.B. constructed the flow systems. E.H.H., A.T. and H.H. conducted the modelling work. T.B., E.H.H. and A.T. wrote the draft and all authors contributed to the final version of the manuscript.

## Funding

## Competing interests

The authors declare no competing interests.
