## [Peer Review File · Nature Communications]

nature portfolio

Peer Review File

Direct sulfuric acid formation from the gas-phase oxidation of reduced-sulfur compoundsREVIEWER COMMENTS

Reviewer #1 (Remarks to the Author):

The manuscript experimentally demonstrated the direct formation of H₂SO₄ in the oxidation of reduced-sulfur compounds. The rate limiting steps for H₂SO₄ production under low-NO_x condition were determined to be CH₃SO₂ and CH₃SO₂OO+RO₂. Besides, the impact of O₃ concentration, RO₂, NO, and NO₂ were investigated. The direct formation of H₂SO₄ from these sulfur compounds over the oceans is very important. And the results are very interesting. The logical of the whole manuscript is not very clear, which limited readers to understand the main mechanism. The manuscript need to be revised then can be considered to publish.

Comments:

- 1.The formation of CH₃SO₂ is the most important intermediate to support the direct formation of H₂SO₄. However, as the authors stated that in line 83: for CH₃SO₂, contributions from CH₃SOO or from CH₃SO₂OO, how to confirm the formation of CH₃SO₂? Can you give some other methods to confirm it?
- 2.Two flow systems, i.e., in the free-jet flow system and the laminar flow tube (LFT) were carried out during the experiments. Please give the reason to use two flow system, what new information can be deduced combined different systems?
- 3.Both of the two part of "Detectable products from CH₃S oxidation" and "Formation routes to direct H₂SO₄" described the oxidation mechanism of these sulfur compounds. They have some connection, and can't be distinguish clearly. Besides, these parts also have the closely linked with the methods part and can get better understanding the mechanism. I suggested the authors to revise these two part to give more clear explanations of the mechanism.
- 4.How were different NO_x conditions considered in the model? The manuscript stated that significant relevance of the direct H₂SO₄ route only exists for SO₂/DMS<5.3, and an air mass change from anthropogenically influenced SO₂/DMS>10. How about the concentration of NO_x for SO₂/DMS<5.3?

Reviewer #2 (Remarks to the Author):

This study builds on recent advancements investigating the oxidation chemistry of organic reduced sulfur compounds in atmospheric conditions. The authors find that DMS, MeSH, and DMDS can be oxidized to H₂SO₄ directly through the intermediates CH₃SO₂, CH₃SO₃, and CH₃SO₂OO. It is surprising that oxidation via pathways involving these intermediate compounds could form up to 50% of sulfuric acid in pristine marine conditions. This study is a helpful advancement to understanding DMS oxidation in the atmosphere with implications for climate.

Evaluation of this study requires expertise in API-TOF mass spectrometry, DMS oxidation chemistry, and atmospheric chemistry modeling. I have expertise in the two latter areas, although I do not use the MCM model. Another reviewer should provide feedback on API-TOF measurements.

Overall, I believe this paper could be appropriate for publication in Nature Communications, but I have questions about the modeling results and implications shown in Figure 4b. I believe that Figure 4b is supposed to represent modeling of pristine marine conditions, and part of the misunderstanding could arise if it is only representing Baring Head, New Zealand. Either way, this paper does not discuss HPMTF formation and subsequent oxidation as a significant source of SO₄ in marine environments. Figure 4b shows the abstraction channel as contributing <5% of SO₄ from DMS. I find this hard to believe based on observations and modelling by Novak et al. (2021), Ye et al. (2022), and Fung et al.

(2022). It is possible that I am missing something, but if I am then I think other readers will also be confused. This question is discussed in more detail below, along with other minor comments and clarifications.

Minor clarifications and comments

- In general, I am confused by the phrase "direct H₂SO₄ formation." Does this refer to DMS/MeSH/DMDS that is oxidized to H₂SO₄ without producing SO₂ as an intermediate? Based on Figure 3a, showing that CH₃SH and CH₃SSCH₃ are oxidized by O₃ through pathways that produce SO₂ (pathway 3 and 6a), "direct H₂SO₄ formation" includes H₂SO₄ formed through the oxidation of reduced sulfur compounds that includes SO₂ as an intermediate. So what is the difference between "H₂SO₄ formation" and "direct H₂SO₄ formation?"
- Throughout the text, the author states that SO₂ oxidation to SO₄ proceeds only via OH and Criegee Intermediates. This statement does not include SO₂ oxidized by H₂O₂, O₃, O₂ catalyzed by transition metals such as iron and manganese, isoprene hydroxyl peroxides, and by reactive halogens such as HOBr and HOCl. There is substantial evidence that the mechanisms not mentioned by the authors could account for a large portion of SO₂ oxidation globally. See, for example, Alexander et al. (2012), Chen et al. (2017), and Dovrou et al. (2019). Does MCM/CAPRAM have SO₂ oxidation via these pathways? I assume that it does. If those reactions are included in MCM/CAPRAM, I suggest that the authors change all figures showing "SO₂ + OH" to say "SO₂ + oxidants" and update the text to remove this oversimplification
- The authors investigate organic reduced sulfur compound oxidation by observing intermediates and products from the oxidation of CH₃SH and CH₃SSCH₃. These starting reactants produce CH₃S and CH₃SO₂ as the initial intermediate radicals. Neither of these intermediate radicals lead to the formation of HPMTF (HOCH₂SCHO), which has been observed as a major intermediate oxidation product of DMS that is globally ubiquitous in the marine boundary layer (Novak et al., 2021; Veres et al., 2021; Ye et al., 2022). I understand that CH₃SH and CH₃SSCH₃ were used as starting reactants to avoid complicated DMS + OH chemistry. However, it seems a strange omission in the article to not mention HPMTF as a significant pathway for H₂SO₄ formation in marine environments. I see in Supplementary Table 5 that HPMTF is included in the MCM mechanism abstraction channel shown in figure 4b, but I don't see a reaction converting HPMTF into H₂SO₄, which is included in recent global chemical transport modeling (e.g. Novak et al., 2021; Fung et al. 2022). I find it very surprising that the abstraction channel, which includes HPMTF, contributes only <1% - 3% of in-cloud H₂SO₄ production in Figure 4b. This is hard to believe based on observations and modelling by Ye et al. (2022) and Fung et al. (2022). Perhaps I am missing something, I apologize if this is the case. Either way, I think the authors should clarify why the abstraction pathway is so insignificant in their model results compared to other studies.
- Writing suggestion: in the paragraph starting with "It is remarkable that H₂SO₄ and CH₃SO₃H concentrations almost uniformly increased..." the first sentence implies that step 20 can compete with step 17. However, in the following sentence, they state that CH₃SO₃ mostly likely abstracts the H atom from CH₃SH to form CH₃SO₃H, rather than CH₃SO₃ reacting with HO₂. I think the authors should clarify this paragraph to make sure they are not implying that step 20 is not actually competing with step 17. Specifically, rather than the word "entail" in the second sentence, perhaps use the word "imply" to make it clear that this reaction is not actually occurring.
- Writing suggestion: in the paragraph starting with "Experiments with varying ozone concentration", the first sentence implies that CH₃SO₂ + O₃ increases with increasing O₃, but the actual result is that only a small portion proceeds via this pathway. I recommend structuring this paragraph differently to make it clear that this is unexpected and a new discovery. For example, "Experiments with varying ozone concentration showed increasing H₂SO₄ yields with rising ozone. Experiments with heavy ozone showed that CH₃S oxidation

mainly proceeds by pathways 3 and 6a. This result counters expectations based on..."

- Writing suggestion: in the paragraph starting with "adjustments in the H₂SO₄ yields," please change the word "being" in the following sentence to "to be:" "Adjusted H₂SO₄ yields for low-NO_x conditions and [O₃] = 5.7 +/- 1011 molecules cm⁻³ were estimated being 0.074 +/- 0.015 % per formed CH₃S and 0.82 +/- 0.02 % per formed CH₃SO₂ (Fig.2a) assuming a CH₃SO₂ yield of unity from OH + CH₃SSCH₃."
- Writing suggestion: this paper might be easier to follow for a broader scientific audience, especially the atmospheric chemistry modeling community, if acronyms are used for some compounds, such as MeSH for CH₃SH, DMDS for CH₃SSCH₃, and MSA for CH₃SO₃H.
- Question on potential scientific implication: the authors find that the reaction CH₃SO₃ + HO₂  CH₃SO₃H does not occur in their experiments. It would be useful to know how CH₃SO₃H forms in the presence of high NO, as shown in Figure 2a, which seems similar to what is observed in Ye et al. (2022) Figure 2a. Modeling in Ye et al. (2022) (Figure 2c) could not reproduce observed concentrations of MSA. Perhaps the experiments presented in this study could help answer this question. It's possible that the differences in the studies makes it difficult to answer this question (e.g. Ye et al., 2022 used DMS instead of CH₃SH and CH₃SSCH₃ used in this study).

In summary, I think this paper presents an advancement in organic reduced sulfur compound oxidation chemistry and I would like to see it published. My main request is that the authors clarify why their modeling results, especially the summary presented in Figure 4b, include so little H₂SO₄ produced via DMS  HPMTF  SO₄²⁻, and why their results differ in the relevance of the abstraction pathway compared to modeling by others (Novak et al. 2021, Fung et al. 2022, Ye et al. 2022). I think it would also be wise to at least mention relevant sulfur compounds and chemistry that has been oversimplified in the text, such as SO₂ oxidation proceeding via many pathways (OH, O₃, H₂O₂, HOBr, HOCl, ISOPOOH, TMI-O₂) and HPMTF as a significant intermediate compound in organic reduced sulfur compound oxidation.

References cited in this review:

- Alexander B, Allman D J, Amos H M, Fairlie T D, Dachs J, Hegg D A and Sletten R S 2012 Isotopic constraints on the formation pathways of sulfate aerosol in the marine boundary layer of the subtropical northeast Atlantic Ocean *Journal of Geophysical Research Atmospheres* 117
- Chen Q, Schmidt J A, Shah V, Zaegle L, Sherwen T and Alexander B 2017 Sulfate production by reactive bromine: Implications for the global sulfur and reactive bromine budgets *Geophys Res Lett* 44 7069-78
- Dovrou E, Rivera-Rios J C, Bates K H and Keutsch F N 2019 Sulfate Formation via Cloud Processing from Isoprene Hydroxyl Hydroperoxides (ISOPOOH) *Environ Sci Technol* 53 12476-84
- Fung K M, Heald C L, Kroll J H, Wang S, Jo D S, Gettelman A, Lu Z, Liu X, Zaveri R A, Apel E C, Blake D R, Jimenez J L, Campuzano-Jost P, Veres P R, Bates T S, Shilling J E and Zawadowicz M 2022 Exploring dimethyl sulfide (DMS) oxidation and implications for global aerosol radiative forcing *Atmos Chem Phys* 22 1549-73
- Novak G A, Fite C H, Holmes C D, Veres P R, Neuman J A, Faloon I, Thornton J A, Wolfe G M, Vermeuel M P, Jernigan C M, Peischl J, Ryerson T B, Thompson C R, Bourgeois I, Warneke C, Gkatzelis G I, Coggon M M, Sekimoto K, Bui T P, Dean-Day J, Diskin G S, Digangi J P, Nowak J B, Moore R H, Wiggins E B, Winstead E L, Robinson C, Lee Thornhill K, Sanchez K J, Hall S R, Ullmann K, Dollner M, Weinzierl B, Blake D R and Bertram T H 2021 Rapid cloud removal of dimethyl sulfide oxidation products limits SO₂ and cloud condensation nuclei production in the marine atmosphere *Proceedings of the National Academy of Sciences* 118 e2110472118 Online:
<https://doi.org/10.1073/pnas.2110472118>
- Veres P R, Andrew Neuman J, Bertram T H, Assaf E, Wolfe G M, Williamson C J, Weinzierl B,

Tilmes S, Thompson C R, Thames A B, Schroder J C, Saiz-Lopez A, Rollins A W, Roberts J M, Price D, Peischl J, Nault B A, Moller K H, Miller D O, Meinardi S, Li Q, Lamarque J F, Kupc A, Kjaergaard H G, Kinnison D, Jimenez J L, Jernigan C M, Hornbrook R S, Hills A, Dollner M, Day D A, Cuevas C A, Campuzano-Jost P, Burkholder J B, Paul Bui T, Brune W H, Brown S S, Brock C A, Bourgeois I, Blake D R, Apel E C and Ryerson T B 2020 Global airborne sampling reveals a previously unobserved dimethyl sulfide oxidation mechanism in the marine atmosphere Proc Natl Acad Sci U S A 117 4505-10

Ye Q, Goss M B, Krechmer J E, Majluf F, Zaytsev A, Li Y, Roscioli J R, Canagaratna M, Keutsch F N, Heald C L and Kroll J H 2022 Product distribution, kinetics, and aerosol formation from the OH oxidation of dimethyl sulfide under different RO₂ regimes Atmos Chem Phys 22 16003-15 Online: <https://acp.copernicus.org/articles/22/16003/2022/>

Reviewer #3 (Remarks to the Author):

The authors report the observation of direct sulfuric acid formation from the oxidation of organic reduced-sulfur compounds which are normally emitted globally from natural sources over the ocean. They provide experimental evidence and implement newly observed reactions for model comparison to experimental findings. The sulfuric acid yield from these direct pathways were then adjusted to make it compatible with real atmospheric conditions. Finally, model simulations were performed to analyze the impact of this direct source of sulfuric acid from oceanic emission of reduced sulfur sources compared to traditional pathway of sulfuric acid formation (oxidation of OH+SO₂). The conclusions for importance of this direct route in the Southern hemisphere under pristine marine condition and the importance of considering loss routes via clouds chemistry is notable.

The paper is well written, well structured, and broadly covers some questions addressing knowledge gaps relevant to recent updates in marine sulfur chemistry the community has had since recent lab, field, and modeling work on novel atmospheric sulfur chemistry relevant to marine emission and their contribution for new particle formation. I would recommend publication after considering the minor points and suggestions below.

Line 68: While it is of course mentioned it in the main text, I think visually it would be good to include the formula of methylthiol, dimethyl sulfide and dimethyl disulfide in the figure caption along with the names, that way a reader can easily related the scheme and the figure caption.

Figure 4 - I suggest reversing the color bar of the legend for Figure 4(a) so that darker color represents higher values of average DMS concentration.

Supplementary Table 1 - Consider condensing the experiment description for column 1. Either rename the column or put the relative humidity value in a separate column. For column 4, I suggest renaming as 'Ionization technique' with entries as 'Nitrate/Iodide' for simplicity.

Supplementary Table 3 - Instead of just mentioning 'Averaged net reaction rates of daytime DMS, CH₃SH and SO₂ oxidation (in molecules cm⁻³s⁻¹) over the period between the second to the fourth model day of the different simulations (in red)' in the figure caption add 'Day 2-4 average rate' for the red values in the body of the table. The manuscript does have lots of figures and tables and most of them have very descriptive titles, so more clarification/specification in the organization can enhance the clarify to the readers.

Supplementary Table 5 - Formation of HOOCH₂SCHO is considered as an important step for atmospheric DMS oxidation from recent observation and experimental studies. For CAPRAM-DM1.0, I find that photolysis of HOOCH₂SCHO was considered. But how does chemical loss of HOOCH₂SCHO by OH or it's cloud uptake do you think would contribute to the major findings of this present work? Unless I missed something to correlate, despite of ignoring reactions due to uncertainty in reaction kinetics, it would be great if it's possible to include a range of expected outcomes for such processes.

General comment: Is it possible to evaluate the aerosol number concentrations with this additional source of direct sulfuric acid formation using the model used in this study.

A point-by-point response to the comments on the manuscript:
Nature Communications NCOMMS-23-09514-T

Direct sulfuric acid formation from the oxidation of reduced-sulfur compounds

REVIEWER COMMENTS

Reviewer #1 (Remarks to the Author):

The manuscript experimentally demonstrated the direct formation of H₂SO₄ in the oxidation of reduced-sulfur compounds. The rate limiting steps for H₂SO₄ production under low-NO_x condition were determined to be CH₃SO₂ and CH₃SO₂OO+RO₂. Besides, the impact of O₃ concentration, RO₂, NO, and NO₂ were investigated. The direct formation of H₂SO₄ from these sulfur compounds over the oceans is very important. And the results are very interesting. The logical of the whole manuscript is not very clear, which limited readers to understand the main mechanism. The manuscript need to be revised then can be considered to publish.

We thank the reviewer for the positive feedback and the recommendation to consider our manuscript for publication in Nature Communications after addressing the comments. A point-by-point answer to your comments is provided below for every specific comment.

With regards to your point of "*The logical of the whole manuscript is not very clear, which limited readers to understand the main mechanism.*", we want to address at this point again that we originally submitted the manuscript to Nature where we got the opportunity to transfer the manuscript directly to Nature Communications. Therefore, the original manuscript had less descriptions and discussions in the main body due to the length restrictions. The format and manuscript size of Nature Communications is different. Thus, the revised manuscript includes several changes. We have changed slightly the structure of the manuscript, for example we moved text from the methods sections to the results parts, as we think that it fits there better. Moreover, we added more subheadings, explanations and discussions for more clarity and a better readability of the manuscript, as Nature Communications provides us more space for that. We would like to thank the reviewer again for bringing this issue to our attention. For details on how the whole manuscript has been changed, please see the manuscript with tracked changes.

Comments:

R1C1: The formation of CH₃SO₂ is the most important intermediate to support the direct formation of H₂SO₄. However, as the authors stated that in line 83: for CH₃SO₂, contributions from CH₃SOO or from CH₃SO₂OO, how to confirm the formation of CH₃SO₂? Can you give some other methods to confirm it?

The direct observation of CH₃ SO₂ for close to atmospheric conditions is very challenging. To the best of our knowledge, it is not reported so far in the literature. We stated that the signal

attributed to CH₃SO₂ could be influenced by contributions from CH₃SOO or CH₃SO₂O₂ after fragmentation. Because fragmentation after clustering with iodide is highly unlikely, we have removed the latter option in the revised version. However, it is impossible to distinguish between CH₃SO₂ and CH₃SOO with our mass spectrometric technique. Thus, we have to consider this option. A more promising way for qualitative investigation of sulfur oxidation are cryogenic techniques using spectroscopic methods for product identification.

The following changes/additions have been done incl. three new references, 36, 37 and 38:

"For the latter, a contribution from CH₃SOO cannot be ruled out."

"Basically, the direct observation of CH₃SO₂ and other intermediates of the CH₃S oxidation for close to atmospheric conditions appears to be very challenging^{36,37}. A spectroscopic study on the product formation of CH₃S + O₂ in cryogenic matrixes unambiguously identified CH₃SOO, CH₃SO₂ and CH₃SO₂O₂ as important intermediates supporting the relevance of the reaction sequence 1/-1, 2 and 11/-11 (Fig. 1) in the CH₃S oxidation¹⁵. Cryogenic matrix techniques in general represent an useful approach for qualitative studying of sulfur oxidation³⁸."

R1C2: Two flow systems, i.e., in the free jet flow system and the laminar flow tube (LFT) were carried out during the experiments. Please give the reason to use two flow system, what new information can be deduced combined different systems?

The flow tubes allow experiments with different residence times, i.e., 7.9 and 32 s. Starting with the free-jet flow system, we realized that 7.9 s are probably not long enough for efficient CH₃SO₃ decomposition, which was later confirmed in experiments with 32 s. Based on the results from both flow tubes, we got an estimate of the rate coefficient $k_{\text{(CH}_3\text{SO}_3 \text{ decomposition)}}$. This is mentioned now in Introduction and Methods:

"Here we experimentally demonstrate the direct formation of H₂SO₄ from the OH radical-initiated gas-phase oxidation of organic sulfur compounds by its direct mass spectrometric detection in two flow systems²⁹⁻³¹ under atmospheric conditions with residence times of 7.9 and 32 s. ..

"The flow tubes worked with different residence times, 7.9 and 32 s, respectively, that allowed to draw a conclusion regarding the rate of relatively slow processes, here on the thermal decomposition of CH₃SO₃."

R1C3: Both of the two part of "Detectable products from CH₃S oxidation" and "Formation routes to direct H₂SO₄" described the oxidation mechanism of these sulfur compounds. They have some connection, and can't be distinguish clearly. Besides, these parts also have the closely linked with the methods part and can get better understanding the mechanism. I suggested the authors to revise these two parts to give more clear explanations of the mechanism.

The paragraph "Detectable products from CH₃S oxidation" should simply show what's possible to measure with our experiment and an example of an overview experiment is given. Now it is also stated that SO₂ represents the main product, as already published earlier, and we are focusing here only on the non-SO₂ products.

"Product ionization by means of iodide (I⁻) and nitrate (NO₃⁻) in the mass spectrometric analysis was found as a suitable way to observe product formation, other than SO₂, in the oxidation process. Recently, an experimental SO₂ yield of 86 ± 18 % has been reported for low-NO conditions qualifying SO₂ as the predominant product^{32,33}. Fig. 2a shows the detected products, other than SO₂, from an overview experiment on the oxidation of CH₃S initiated by the reaction $\text{OH} + \text{CH}_3\text{SH} \longrightarrow \text{CH}_3\text{S} + \text{H}_2\text{O}$ ^{33,34} (Fig. 1)."

The MSA part was removed from "Detectable products from CH₃S oxidation", it's now the new paragraph "MSA formation induced by elevated CH₃SH and DMDS concentrations" with more detailed explanations. We added also a figure from Extended Data Figures (former Extended Data Fig. 4) into the main text that supports the mechanistic discussion on MSA formation.

"MSA formation induced by elevated CH₃SH and DMDS concentrations

It is remarkable that H₂SO₄ and MSA concentrations increased almost uniformly with rising CH₃SH conversion, which was accompanied by a rising HO₂ radical level for the chosen reaction conditions (Fig. 2a and Supplementary Fig. 2). The competing steps **17** vs. **20** entail a decreasing H₂SO₄ / MSA ratio with rising HO₂ concentrations (Fig. 1), which is not visible in the experiments (Supplementary Fig. 3). This implies that our experimental findings do not support considerable MSA formation via CH₃SO₃ + HO₂ (pathway **20**). Moreover, an increasing H₂SO₄ / MSA ratio with decreasing CH₃SH concentration was observed for otherwise nearly constant reaction conditions, including CH₃SH consumption by the OH reaction and the HO₂ concentration (Fig. 3). The MSA signal practically disappeared for CH₃SH concentrations below a few 10^{10} molecules cm⁻³. Thus, the reaction of CH₃SO₃ with CH₃SH (pathway **19**), likely via H-abstraction of the labile S-bound H atom, seems to dominate the MSA formation under the present conditions. This also means that the direct H₂SO₄ formation via CH₃SO₃ decomposition (pathway **17**) is suppressed in the presence of sufficiently high CH₃SH or other substances serving as H-atom donor. A similar behaviour of the H₂SO₄ / MSA ratio was also observed in OH + DMDS experiments varying the DMDS concentrations (Supplementary Fig. 4). Here, almost exclusive H₂SO₄ formation can be expected for DMDS concentrations below 10^{10} molecules cm⁻³.

Because atmospheric CH₃SH and DMDS concentrations are clearly smaller than 10^{10} molecules cm⁻³ (**400** ppt), see attached MS Excel spreadsheet and ref.⁵, CH₃SO₃ decomposition (pathway **17**) forming finally the direct H₂SO₄ most likely dominates the fate of CH₃SO₃ for atmospheric conditions (Fig. 3 and Supplementary Fig. 4). MSA formation according to CH₃SO₃ + RH for RH = CH₃SH or DMDS (pathway **19**) has to be of minor importance. It is speculative whether another hydrocarbons RH could efficiently form MSA via pathway **19** in the atmosphere or not.

Fig. 3: Concentrations of H₂SO₄ and MSA and the MSA / H₂SO₄ ratio as a function of CH₃SH concentration. Experiments on OH + CH₃SH were carried out in the free jet flow system at r.h. = 10% and a reaction time of 7.9 s using IPN photolysis for OH radical generation. Reactant concentrations are stated in Supplementary Table 1. The error bars for H₂SO₄ and MSA depict the uncertainty of -20% based on the uncertainty in the calibration factor.

Now, the two paragraphs "MSA formation induced by elevated CH₃SH and DMDS concentrations" and "Formation routes to direct H₂SO₄" comprise all mechanistic information regarding MSA and H₂SO₄ formation, respectively. All mechanistic information from the former Method section is given now in the main body.

We hope, that we thereby clarified the structure of the manuscript and improved the readability.

Below we provide the main changes in the paragraph "Formation routes to direct H₂SO₄":

"Ozone: No significant H₂SO₄ formation from OH + CH₃SH was observed for ozone concentrations of up to 2×10^{12} molecules cm⁻³ (—80 ppb) in the free jet flow system with the short reaction time of 7.9 s. H₂SO₄ became detectable in the laminar flow tube (LFT) with a reaction time of 32 s indicating a relatively slow process of direct H₂SO₄ formation (Fig. 4a). Big differences in the H₂SO₄ yields of more than an order of magnitude were measured using either OH + CH₃SH for CH₃S generation or OH + DMDS forming CH₃S and most likely CH₃SO₂ with high yields. Considering CH₃SO₂ as the needed intermediate for H₂SO₄ formation (Fig. 1), CH₃S's oxidation obviously proceeds only with a small share via CH₃SO₂, e.g., $\leq 9\%$ for an ozone concentration of 5.7×10^{11} molecules cm⁻³ (Fig. 4a) taking OH + DMDS with a CH₃SO₂ yield of unity as the reference. Moreover, OH + CH₃SH experiments with heavy ozone (¹⁸O₃) revealed the absence of H₂SO₄ containing three ¹⁸O atoms (Supplementary Fig. 5) as expected from the reaction sequence **3**, **6b** and **9** (Fig. 1). We largely measured H₂SO₄ with one ¹⁸O atom consistent with the reaction sequence **1/-1**, **2** and

9. The findings imply the dominance of pathway **6a** over **6b** or in fact the irrelevance of pathway **6b**, allowing for the importance of ozone reactions in the CH₃S oxidation^{8,31}. This can be explained by the high exothermicity of the CH₃SO + O₃ reaction forming the chemically excited CH₃SO₂* that rapidly decomposes to SO₂ and CH₃ before it is thermalised⁴².

The small H₂SO₄ yields < 1% for atmospheric ozone concentrations, even under conditions of the preferred CH₃SO₂ generation from OH + DMDS, support the efficient decomposition of CH₃SO₂ (pathway **8**), which is in line with the high SO₂ yields reported recently^{32,33}. SO₃ yields measured under dry conditions, r.h. < 0.1%, were in very good agreement with the H₂SO₄ yields at r.h. = 10% (Fig. 4a) confirming SO₃ as the precursor of H₂SO₄ (pathways **17** and **18**).

An ozone concentration of 5.7 x 10¹¹ molecules cm⁻³ (-23 ppb) was chosen in the following experiments, which stands for an average concentration over pristine oceans⁴³, making our findings applicable to the atmospheric reaction system.

RO₂ radicals: We detected a distinct impact of RO₂ radicals on the formation of H₂SO₄ and MSA (Fig. 4b). Main RO₂ radicals in the reaction system are CH₃C(O)CH₂O₂, formed in the course of OH generation via THE ozonolysis^{44,45}, and CH₃O₂ as the by-product of SO₂ in the oxidation of CH₃SH and DMDS^{32,33} as well as from OH + CH₄ in the case of CH₄ additions. In the OH + CH₃SH reaction, we increased in a two-step process the concentrations of CH₃C(O)CH₂O₂ and CH₃O₂ radicals, first by a factor of -10, i.e., from 6.2 x 10⁷ to 5.7 x 10⁸ and from 4.0 x 10⁷ to 4.4 x 10⁸ molecules cm⁻³, respectively, leading to enhanced H₂SO₄ formation by a factor of -3.5 for constant CH₃SH conversion (Fig. 4b). Further doubling of the RO₂ concentrations led to further rise in H₂SO₄ productions. The MSA formation, however, increased stronger than that of H₂SO₄, which became clearer visible from a similar experiment on OH + DMDS (Supplementary Fig. 6). Furthermore, we observed predominate MSA formation in a reaction system with HO-C₆H₁₂O₂ along with CH₃C(O)CH₂O₂ as the main RO₂ radicals (Supplementary Fig. 7). It can be speculated that most likely CH₃SO₂OO reacted with RO₂ radicals either via the alkoxy channel (pathway **15**), forming finally H₂SO₄, or via the dismutation channel (pathway **16**), similar to the well-known chemistry of carbon-centred RO₂ radicals⁴⁶, leading to MSA. The branching ratio of pathways **15** vs. **16** appears to be dependent on the structure of the reacting RO₂ radical. Other RO₂ driven pathways, influencing the product formation, cannot be ruled out.

NO: Addition of NO substantially accelerated the H₂SO₄ formation in all experiments (Fig. 4c) supporting the potential importance of CH₃SO₂OO for H₂SO₄ formation, here via pathway **14**. An increase in the H₂SO₄ production by a factor of —4 (Supplementary Fig. 8) was measured using a NO concentration of 1 x 10⁹ molecules cm⁻³ similar to the behaviour observed for elevated RO₂ concentrations (Fig. 4b and Supplementary Fig. 6). This indicates rate coefficients k₁₄ and k₁₅ for the reaction of CH₃SO₂OO with NO and RO₂, respectively, being in the same range. Comparison of results for relatively low NO concentrations of < 10¹⁰ molecules cm⁻³ in the LFT showed more than an order of magnitude higher H₂SO₄ yields from the oxidation of DMDS relative to CH₃SH, in line with the findings from the pure ozone-driven reaction (Fig. 4b). For elevated NO levels, other NO reactions presumably disturbed the CH₃SO₂ formation from OH + DMDS inhibiting further rise of the H₂SO₄ yield. Further NO reactions in the CH₃S oxidation could also negatively impact the H₂SO₄

formation, such as $\text{CH}_3\text{S} + \text{NO} \rightarrow \text{CH}_3\text{SNO}^{\text{10}}$ or $\text{CH}_3\text{SOO} + \text{NO} \rightarrow \text{CH}_3\text{SO} + \text{NO}_2^{\text{10}}$ (pathway **5**) resulting finally in SO_2 formation via pathway **6a**. The higher H_2SO_4 yields from $\text{OH} + \text{CH}_3\text{SH}$ measured in the LFT at $t = 32$ s point again to a slow process of H_2SO_4 formation that is far away from completeness for the reaction time of 7.9 s in the free jet flow system.

NO₂: Addition of NO_2 featured a similar effect for the rise of H_2SO_4 yields (Fig. 4d) as observed for NO (Fig. 4c), albeit the NO_2 impact was less pronounced. It is supposed in the literature that NO_2 reacts with CH_3SO_2 forming CH_3SO_3 (pathway **10**) which finally leads to H_2SO_4 analogous to the ozone-mediated route (pathway **9**)⁹. This set of experiments confirmed again the much higher potential of H_2SO_4 formation starting from $\text{OH} + \text{DMDS}$ regarding $\text{OH} + \text{CH}_3\text{SH}$ as well as the slow formation rate of the direct H_2SO_4 production. H_2SO_4 production almost doubled as the result of a NO_2 addition of 6.7×10^9 molecules cm^{-3} in the LFT experiments (Fig. 4d) indicating nearly the same reaction rate in the reaction of CH_3SO_2 with ozone and NO_2 (pathways **9** and **10**), $[\text{O}_3] = 5.7 \times 10^{11}$ molecules cm^{-3} . This leads to $k_9 / k_{10} \sim 1/85$ being in good agreement with the rate coefficient ratio currently used in models^{21,22}. The experiments with NO_2 addition did not allow any conclusions regarding the relative importance of the product channels **7a** and **7b** from $\text{CH}_3\text{SO} + \text{NO}_2$."

R1C4: How were different NO_x conditions considered in the model? The manuscript stated that significant relevance of the direct H_2SO_4 route only exists for $\text{SO}_2/\text{DMS} < 0.3$, and an air mass change from anthropogenically influenced $\text{SO}_2/\text{DMS} > 10$. How about the concentration of NO_x for $\text{SO}_2/\text{DMS} < 0.3$?

Changes in the NO_x conditions can be considered through the emission rate of NO implemented in the model. The default NO emission rate is set to be 2.5×10^3 molecules $\text{cm}^{-3} \text{ s}^{-1}$. This emission rate is low, but representative for pristine marine conditions in the Southern Ocean. With this NO emission rate in the default cases, average mixing ratios of 2 ppt NO_x are modelled. The SO_2/DMS ratios are below 0.3 for the simulation with cloud occurrence and between 0.3 and 0.8 for simulations without clouds included (see Supplementary Figure 11). However, in pristine regions of the Northern hemisphere, mixing ratios of a few tens of ppt are typically measured, e.g. Cape Verde (Reed et al., 2017). To take this higher concentrations of NO_x into account, we have performed additional sensitivity simulations with an increased NO emission rate (factor of ten higher) that results into average NO_x mixing ratio of about 15 ppt. Interestingly, the increased NO_x concentrations have only a small effect on the yields of primarily formed CH_3S and more or less no effect on the yields of primarily formed CH_3SO_2 in either simulation with or without cloud passages of the air parcel (see Supplementary Table 3 for details).

In the original manuscript submitted to Nature, we included simulations with low NO emission, only. Due to the lower restrictions of the manuscript size of Nature Communications, we included the results of the higher NO_x simulations (see Supplementary Table 3) as well as the modelled NO data (see Supplementary Figure 12). We hope this provides the reader a more comprehensive picture.

Supplementary Table 3: Day 2-4 averaged rates of daytime DMS, CH₃SH and SO₂ oxidation (red) and primary daytime production rates of CH₃S and CH₃SO₂ (blue) of the different simulations. The **bold** values refer towards the molar yields from DMS or CH₃SH oxidation, respectively. Normalized reaction rates ($k_{i,ST}$) are calculated from the averaged net reaction rates divided by the modeled concentrations of DMS, CH₃SH and SO₂, respectively.

	DMS Oxidation		CH ₃ SH Oxidation	OH + SO ₂
	Addition / molec. cm ⁻³ s ^t	Abstraction / molec. cm ⁻³ s ⁻¹	molec. cm⁻³ s¹	molec. cm ⁻³ s ^t
Cloud				
Total daytime 2-4 average rate (avg. kiss)	4.8x10 ⁴ (7.9x10 ⁻⁶ S ⁻¹)	1.9x10 ⁴ (3.3x10 ⁻⁶ S ⁻¹)	8.0x10 ³ (1.6x10 ⁻⁵ s ⁻¹)	7.0x10 ¹ (1.6x10 ⁻⁶ s ¹)
Daytime 2-4 average rate of primarily formed CH ₃ S		4.1x10 ³ 22%	8.0x10 ³ 100%	
Daytime 2-4 average rate of primarily formed CH ₃ SO ₂	2.1x10 ³ 4%			
Cloud with ten times higher NO emission rate				
Total daytime 2-4 average rate (avg. kiss)	4.8x10 ⁴ (8.8x10 ⁻⁶ S ⁻¹)	2.2x10 ⁴ (4.1x10 ⁻⁶ S ⁻¹)	8.2x10 ³ (1.8x10⁻⁵ S⁻¹)	9.2x10 ¹ (2.1x10 ⁻⁶ s ⁴)
Daytime 2-4 average rate of primarily formed CH ₃ S		4.1x10 ³ 19%	8.2x10 ³ 100%	
Daytime 2-4 average rate of primarily formed CH ₃ SO ₂	2.4x10 ³ 5%			
Cloud, lower HA (HA, 298K from Wollesen de Jonge et al., 2021)¹				
Total daytime 2-4 average rate (avg. kiss)	4.8x10 ⁴ (7.9x10 ⁻⁶ s ^t)	1.9x10 ⁴ (3.3x10 ⁻⁶ s ^t)	8.0x10 ³ (1.6x10 ⁻⁵ s ^t)	7.4x10 ¹ (1.2x10 ⁻⁶ S ⁻¹)

Daytime 2-4 average rate of primarily formed CH ₃ S		4.2x10 ³ 22%	8.0x10 ³ 100%
---	--	-----------------------------------	------------------------------------

Daytime 2-4 average rate of primarily formed CH ₃ SO ₂	7.3x10 ³ 15%		
---	-----------------------------------	--	--

no Cloud

Total daytime 2-4 average rate (avg. kiss)	5.1x10⁴ (9.2x10 ⁻⁶ s ⁻¹)	2.5x10⁴ (4.5x10 ⁻⁶ s ⁻¹)	8.5x10³ (2.1x10 ⁻⁵ s ⁻¹)	1.2x10³ (4.6x10 ⁻⁷ s ⁻¹)
--	--	--	--	--

Daytime 2-4 average rate of primarily formed CH ₃ S		6.8x10 ³ 27%	8.5x10 ³ 100%
---	--	-----------------------------------	------------------------------------

Daytime 2-4 average rate of primarily formed CH ₃ SO ₂	4.6x10 ³ 9%		
---	----------------------------------	--	--

no Cloud with ten times higher NO emission rate

Total daytime 2-4 average rate (avg. kiss)	4.8x10⁴ (9.7x10 ⁻⁶ s ⁻¹)	2.8x10⁴ (5.7x10 ⁻⁶ s ⁻¹)	8.5x10³ (2.1x10 ⁻⁵ s ⁻¹)	1.4x10³ (5.0x10 ⁻⁷ s ⁻¹)
--	--	--	--	--

Daytime 2-4 average rate of primarily formed CH ₃ S		8.7x10 ³ 31%	8.5x10 ³ 100%
---	--	-----------------------------------	------------------------------------

Daytime 2-4 average rate of primarily formed CH ₃ SO ₂	4.7x10 ³ 10%		
---	-----------------------------------	--	--

no Cloud, lower H_A (H_{A, 298K} from Wollesen de Jonge et al., 2021)¹

Total daytime 2-4 average rate (avg. kiss)	5.1x10⁴ (9.0x10 ⁻⁶ s ⁻¹)	2.4x10⁴ (4.4x10 ⁻⁶ s ⁻¹)	8.4x10³ (2.1x10 ⁻⁵ s ⁻¹)	1.3x10³ (4.4x10 ⁻⁷ s ⁻¹)
--	--	--	--	--

Daytime 2-4 average rate of primarily formed CH ₃ S		6.6x10 ³ 28%	8.4x10 ³ 100%
---	--	-----------------------------------	------------------------------------

Daytime 2-4 average rate of	2.7x10 ⁴ 53%		
-----------------------------------	--	--

Supplementary Fig. 10: Modelling results of DMS and SO₂ mixing ratios. a. Modelled DMS mixing ratio for the different simulations from the second to the fourth model day. **b.** Modelled SO₂ mixing ratio for the different simulations from the second to the fourth model day. The grey shaded bars represent the night time, whereas the light blue bar represents the cloud occurrence when clouds are considered.

Supplementary Fig. 11: Modelling results of the SO_2 / DMS ratio. Modelled SO_2 / DMS concentration ratio for the different simulation cases from the second to the fourth model day. The grey shaded bars represent the night-time, whereas the light blue bar represents the cloud occurrence when clouds are considered.

Supplementary Fig. 12: Modelling results for NOR. Modelled NO mixing ratios for the different simulation cases from the second to the fourth model day, except the ones with lower H_A values, because of similar concentrations and time profiles of the NO mixing ratios in the simulations with lower H_A compared to the H_A Default simulations. The grey shaded bars represent the night-time, whereas the light blue bar represents the cloud occurrence when clouds are considered.

Reviewer #2 (Remarks to the Author):

This study builds on recent advancements investigating the oxidation chemistry of organic reduced sulfur compounds in atmospheric conditions. The authors find that DMS, MeSH, and DMDS can be oxidized to H₂SO₄ directly through the intermediates CH₃SO₂, CH₃SO₃, and CH₃SO₂OO. It is surprising that oxidation via pathways involving these intermediate compounds could form up to 50% of sulfuric acid in pristine marine conditions. This study is a helpful advancement to understanding DMS oxidation in the atmosphere with implications for climate.

Evaluation of this study requires expertise in APi-TOF mass spectrometry, DMS oxidation chemistry, and atmospheric chemistry modeling. I have expertise in the two latter areas, although I do not use the MCM model. Another reviewer should provide feedback on APi-TOF measurements.

Overall, I believe this paper could be appropriate for publication in Nature Communications, but I have questions about the modeling results and implications shown in Figure 4b. I believe that Figure 4b is supposed to represent modeling of pristine marine conditions, and part of the misunderstanding could arise if it is only representing Baring Head, New Zealand. Either way, this paper does not discuss HPMTF formation and subsequent oxidation as a significant source of SO₄ in marine environments. Figure 4b shows the abstraction channel as contributing <5% of SO₄ from DMS. I find this hard to believe based on observations and modelling by Novak et al. (2021), Ye et al. (2022), and Fung et al. (2022). It is possible that I am missing something, but if I am then I think other readers will also be confused. This question is discussed in more detail below, along with other minor comments and clarifications.

We thank the reviewer for the constructive comments. However, we guess that there is a bit misunderstanding of the intention of our manuscript which we think is probably related to the short version of the manuscript (due to length restrictions of Nature, where we first submitted it). The focus of our study is on the pure gas-phase formation of H₂SO₄ formed in a direct way from the oxidation of reduced-sulfur organic compounds, what we call "direct H₂SO₄ formation". With the term "direct H₂SO₄ formation", we do not refer to the gas-phase oxidation of SO₂ initiated by OH radicals, which is unimportant under our conditions, as explicitly mentioned in the revised manuscript.

("It is to be noted here, that H₂SO₄ formation initiated by OH + SO₂ is unimportant under the chosen conditions....")

Since the present study was only focused on the gas-phase H₂SO₄ formation, we did not emphasize on the multiphase oxidation of SO₂ leading to particulate sulfur(VI). Regarding confusion about particulate sulfur(VI), i.e., particulate sulfate (SO₄²⁻), please see the specific comments given below!

Regarding your comment about HPMTF, this compound cannot have an impact on our provided direct gas-phase H₂SO₄ formation, as currently such process is not observed or calculated for it. Further comments on HPMTF are given below (Reviewer#2 Comment#3 — R2C3)

To better clarify the abovementioned issues of the reviewer in the revised version and to avoid any confusion of readers, we added the term "gas-phase" at many places. Please see the revised manuscript with tracked changes.

Minor clarifications and comments

R2C1: In general, I am confused by the phrase "direct H₂SO₄ formation." Does this refer to DMS/MeSH/DMDS that is oxidized to H₂SO₄ without producing SO₂ as an intermediate? Based on Figure 3a, showing that CH₃SH and CH₃SSCH₃ are oxidized by O₃ through pathways that produce SO₂ (pathway 3 and 6a), "direct H₂SO₄ formation" includes H₂SO₄ formed through the oxidation of reduced sulfur compounds that includes SO₂ as an intermediate. So what is the difference between "H₂SO₄ formation" and "direct H₂SO₄ formation?"

We are reporting on the direct gas-phase H₂SO₄ formation from the oxidation of DMS/CH₃SH/DMDS (reduced-sulfur organic compounds) not meaning the oxidation of SO₂ by OH radicals or Criegee intermediates. Although SO₂ formation is a main process, the subsequent SO₂ oxidation under the chosen reaction conditions is less important and does not disturb the direct H₂SO₄ formation. This fact is mentioned now twice in the main body as well as in Methods.

"It is to be noted here, that H₂SO₄ formation initiated by OH + SO₂ is unimportant under the chosen conditions and, thus, H₂SO₄ needs to arise from the CH₃S oxidation directly, see also Methods."

"H₂SO₄ formation starting from the reaction of SO₂ with OH radicals or Criegee intermediates was again small in these measurement series and did not influence the results of direct H₂SO₄ formation significantly, see also Methods."

"Modelling calculations, including the IPN photolysis experiment, confirmed that H₂SO₄ production starting from the reaction of SO₂ with OH radicals or Criegee intermediates did not significantly influence the results of direct H₂SO₄ formation from the organic sulfur compounds."

We added at several places "direct" and "gas-phase" before "H₂SO₄" to make it clearer.

R2C2: Throughout the text, the author states that SO₂ oxidation to SO₄ proceeds only via OH and Criegee Intermediates. This statement does not include SO₂ oxidized by H₂O₂, O₃, O₂ catalyzed by transition metals such as iron and manganese, isoprene hydroxyl peroxides, and by reactive halogens such as HOBr and HOCl. There is substantial evidence that the mechanisms not mentioned by the authors could account for a large portion of SO₂ oxidation globally. See, for example, Alexander et al. (2012), Chen et al. (2017), and Dovrou et al. (2019). Does MCM/CAPRAM have SO₂ oxidation via these pathways? I assume that it does. If those reactions are included in MCM/CAPRAM, I suggest that the authors change all figures showing "SO₂ + OH" to say "SO₂ + oxidants" and update the text to remove this oversimplification

We thank for the suggestion, but we want to clarify again that the present manuscript focuses on the direct gas-phase H₂SO₄ formation only and do not emphasis on the multiphase oxidation of SO₂ leading to particulate sulfur(VI). These are two different compounds, in the view of the

manuscript. For the sake of clarity, in our paper, the term "H₂SO₄" refers to gas-phase sulfuric acid (gas-phase H₂SO₄), but not to particulate sulfate (SO₄²⁻).

The applied aqueous-phase mechanism CAPRAM considers almost all sulfur(VI) formation pathways mentioned above, such as aqueous-phase oxidation of SO₂ by H₂O₂, O₃, transition metal ions, HOCl, HOBr and other oxidants such as HNO₄. Therefore, a comparison between the impact of total S(VI) formation in both gas- and aqueous-phase would be possible. However as known from former model studies (see Tilgner et al. (2021) and references therein), even with the new direct gas-phase H₂SO₄ formation pathway, the aqueous-phase oxidation of dissolved sulfur(IV) (i.e. mainly dissolved SO₂) represents the main sulfur(VI) formation (sulfate (SO₄²⁻)) pathway. Unless the gas-phase sulfur(VI) formation pathway (gas-phase H₂SO₄) is less important for the sulfur(VI) aerosol mass, however, it is a driving factor for other key processes such as new particle formation. As the paper does not focus on multiphase sulfur(VI) aerosol mass formation, a deeper discussion of the aqueous-phase SO₄²⁻ formation pathways was not of the scope of the present study. To better clarify which S(IV) to S(VI) formation pathways are included in the mechanism, we have extended the mechanism description in Methods section. The added text read as follows:

"This mechanism system describes the formation of gas-phase H₂SO₄ and aqueous sulfate in a very detailed manner."

"With these two additional models, CAPRAM4.0 includes almost all known sulfate formation pathways in the atmospheric aqueous phase, such as sulfite oxidation by H₂O₂, O₃, HNO₄, reactive halogen species (X₂ radical or HOX, with X = Cl, Br or I) or transition metal ions."

R2C3: The authors investigate organic reduced sulfur compound oxidation by observing intermediates and products from the oxidation of CH₃SH and CH₃SSCH₃. These starting reactants produce CH₃S and CH₃SO₂ as the initial intermediate radicals. Neither of these intermediate radicals lead to the formation of HPMTF (HOOCH₂SCHO), which has been observed as a major intermediate oxidation product of DMS that is globally ubiquitous in the marine boundary layer (Novak et al., 2021; Veres et al., 2021; Ye et al., 2022). I understand that CH₃SH and CH₃SSCH₃ were used as starting reactants to avoid complicated DMS + OH chemistry. However, it seems a strange omission in the article to not mention HPMTF as a significant pathway for H₂SO₄ formation in marine environments. I see in Supplementary Table 5 that HPMTF is included in the MCM mechanism abstraction channel shown in figure 4b, but I don't see a reaction converting HPMTF into H₂SO₄, which is included in recent global chemical transport modeling (e.g. Novak et al., 2021; Fung et al. 2022). I find it very surprising that the abstraction channel, which includes HPMTF, contributes only <1% - 3% of in-cloud H₂SO₄ production in Figure 4b. This is hard to believe based on observations and modelling by Ye et al. (2022) and Fung et al. (2022). Perhaps I am missing something, I apologize if this is the case. Either way, I think the authors should clarify why the abstraction pathway is so insignificant in their model results compared to other studies.

We thank the reviewer for the hint. However, as written in our answer to R3C5 (see below), the model includes the oxidation of HPMTF by OH in the gas phase producing finally SO₂ (see Supplementary Table 5 for details). In the first manuscript version, we forgot to include it in the Table. We apologize for this and possible confusions to the reader. The HPMTF gas-phase oxidation mechanism included in the mechanism is mainly based on the proposed mechanism by Wu et al. (2015). There, HPMTF oxidation results into OCS or SO₂ formation, but not directly into gas-phase H₂SO₄. Of course, a direct gas-phase formation of H₂SO₄ from subsequent gas-phase HPMTF oxidation might be possible, if similar reaction pathways would occur for HOOCH₂S as considered for CH₃S, but is not verified by any study yet. Therefore, this speculative formation route is not considered in the current chemical mechanism.

The reviewer is right that the mechanistic description of HPMTF oxidation in the atmospheric multiphase is limited, because of omitting aqueous-phase oxidation pathways. Aqueous-phase oxidation of HPMTF definitely occurs, as suggested by the field and laboratory measurements (Veres et al. (2020), Jernigan et al. (2022)), but the pathways are not well investigated and highly uncertain. One is clear, HPMTF oxidation in the aqueous phase results into particulate sulfate (SO₄²⁻) formation, but this is not the topic of the current manuscript. We added a broader description on the implemented HPMTF oxidation pathway that is given below.

"The mechanistic updates comprise the formation of the hydroperoxymethyl thioformate (HPMTF) and its further oxidation in the gas phase. The gas-phase HPMTF oxidation follows mainly the proposed routes described by Wu et al. (2015)¹¹, considering SO₂ or OCS formation, only. Thus, HPMTF cannot contribute to the direct gas-phase formation of H₂SO₄ in this mechanism. Phase transfer and subsequent aqueous-phase processing of HPMTF, not included yet because of the current high uncertainties, do not change this."

Regarding your comment "I find it very surprising that the abstraction channel, which includes HPMTF, contributes only <1% - 3% of in-cloud H₂SO₄ production in Figure 4b. This is hard to believe based on observations and modelling by Ye et al. (2022) and Fung et al. (2022).", we want to mention that HPMTF oxidation contributes more than one quarter to gas-phase SO₂ formation in all simulations performed with our model. The direct gas-phase H₂SO₄ formation through the abstraction channel is a result from the "classical" CH₃SCH₂O₂ chemistry leading to CH₃S.

For your last comment "Either way, I think the authors should clarify why the abstraction pathway is so insignificant in their model results compared to other studies." To answer this comment, we want to address the reviewer to statements in our first mechanistic study on HPMTF formation (Berndt et al., 2019). In this study, we already showed that HPMTF formation dominates the abstraction channel in our simulations and, thus, oxidation of DMS to CH₃S is small (usually < 30%).

The simulations provided in this study indicate that HPMTF lifetime is not determined by gas-phase oxidation. We model a steady increase, pointing towards missing sinks, such as uptake processes. Therefore, the HPMTF contribution to total multiphase S(VI) formation is undervalued. However, a deeper discussion is not in the scope of the current manuscript and thus not done.

R2C4: Writing suggestion: in the paragraph starting with "It is remarkable that H₂SO₄ and CH₃SO₃H concentrations almost uniformly increased..." the first sentence implies that step 20 can compete with step 17. However, in the following sentence, they state that CH₃SO₃ mostly likely abstracts the H atom from CH₃SH to form CH₃SO₃H, rather than CH₃SO₃ reacting with HO₂. I think the authors should clarify this paragraph to make sure they are not implying that step 20 is not actually competing with step 17. Specifically, rather than the word "entail" in the second sentence, perhaps use the word "imply" to make it clear that this reaction is not actually occurring.

Thank you for this hint. We followed your suggestion, it makes it clearer what's meant.

"The competing steps 17 vs. 20 imply a decreasing H₂SO₄ / MSA ratio with rising HO₂ concentrations (Fig. 1), which is not visible in the experiments (Supplementary Fig. 3)."

R2C5: Writing suggestion: in the paragraph starting with "Experiments with varying ozone concentration", the first sentence implies that CH₃SO₂ + O₃ increases with increasing O₃, but the actual result is that only a small portion proceeds via this pathway. I recommend structuring this paragraph differently to make it clear that this is unexpected and a new discovery. For example, "Experiments with varying ozone concentration showed increasing H₂SO₄ yields with rising ozone. Experiments with heavy ozone showed that CH₃S oxidation mainly proceeds by pathways 3 and 6a. This result counters expectations based on..."

It was not fully unexpected that the yield of direct H₂SO₄ starting from CH₃S + O₃ (and subsequent O₃ reactions in this system) was that low. In the literature it was already known from quantum chemistry that CH₃SO + O₃ should mainly form CH₃ and SO₂ rather than CH₃SO₂ due to chemical excitation in CH₃SO₂. So, we should not say "This results counters expectations ...". The point is, that this fact was not experimentally shown before. But the reviewer is right, more explanations are needed. The whole "Ozone" paragraph has been restructured and the explanations from Method are given now in the main text.

"Ozone: No significant H₂SO₄ formation from OH + CH₃SH was observed for ozone concentrations of up to 2 x 10¹² molecules cm⁻³ (—80 ppb) in the free jet flow system with the short reaction time of 7.9 s. H₂SO₄ became detectable in the laminar flow tube (LFT) with a reaction time of 32 s indicating a relatively slow process of direct H₂SO₄ formation (Fig. 4a). Big differences in the H₂SO₄ yields of more than an order of magnitude were measured using either OH + CH₃SH for CH₃S generation or OH + DMDS forming CH₃S and most likely CH₃SO₂ with high yields. Considering CH₃SO₂ as the needed intermediate for direct H₂SO₄ formation (Fig. 1), CH₃S's oxidation obviously proceeds only with a small share via CH₃SO₂, e.g. ≤ 9% for an ozone concentration of 5.7 x 10¹¹ molecules cm⁻³ (Fig. 4a) taking OH + DMDS with a CH₃SO₂ yield of unity as the reference. Moreover, OH + CH₃SH experiments with heavy ozone (¹⁸O₃) revealed the absence of H₂SO₄ containing three ¹⁸O atoms (Supplementary Fig. 5) as expected from the reaction sequence 3, 6b and 9 (Fig. 1). We largely measured H₂SO₄ with one ¹⁸O atom consistent with the reaction sequence 1/-1, 2 and

9. The findings imply the dominance of pathway **6a** over **6b** or in fact the irrelevance of pathway **6b**, allowing for the importance of ozone reactions in the CH₃S oxidation^{8,34,41}. This can be explained by the high exothermicity of the CH₃SO + O₃ reaction forming the chemically excited CH₃SO₂* that rapidly decomposes to SO₂ and CH₃ before it is thermalised⁴²"

R2C6: Writing suggestion: in the paragraph starting with "adjustments in the H₂SO₄ yields," please change the word "being" in the following sentence to "to be:" "Adjusted H₂SO₄ yields for low-NO_x conditions and [O₃] = 5.7 +/- 1011 molecules cm⁻³ were estimated being 0.074 +/- 0.015 % per formed CH₃S and 0.82 +/- 0.02 % per formed CH₃SO₂ (Fig.2a) assuming a CH₃SO₂ yield of unity from OH + CH₃SSCH₃."

We changed this sentence accordingly.

"Adjusted H₂SO₄ yields for low-NO_x conditions and [O₃] = 5.7 x 10¹¹ molecules cm⁻³ were estimated to be 0.074 ± 0.015 % per formed CH₃S and 0.82 ± 0.02 % per formed CH₃SO₂ (Fig. 4a) assuming a CH₃SO₂ yield of unity from OH + DMDS."

R2C7: Writing suggestion: this paper might be easier to follow for a broader scientific audience, especially the atmospheric chemistry modeling community, if acronyms are used for some compounds, such as MeSH for CH₃SH, DMDS for CH₃SSCH₃, and MSA for CH₃SO₃H.

We followed this suggestion and are using now DMS, DMDS and MSA throughout the text and the reaction scheme after introducing the acronyms, but not in the case of methylthiol, where we hold on to CH₃SH. It's not *stylish* for a chemist to write "Me" for a methyl group. Please see the revised manuscript with tracked changes.

R2C8: Question on potential scientific implication: the authors find that the reaction CH₃SO₃ + HO₂  CH₃SO₃H does not occur in their experiments. It would be useful to know how CH₃SO₃H forms in the presence of high NO, as shown in Figure 2a, which seems similar to what is observed in Ye et al. (2022) Figure 2a. Modeling in Ye et al. (2022) (Figure 2c) could not reproduce observed concentrations of MSA. Perhaps the experiments presented in this study could help answer this question. It's possible that the differences in the studies makes it difficult to answer this question (e.g. Ye et al., 2022 used DMS instead of CH₃SH and CH₃SSCH₃ used in this study).

Our study indicates that MSA is mainly formed in the reaction of CH₃SO₃ with CH₃SH or DMDS, and not via CH₃SO₃ + HO₂ as expected so far. The non-relevance of CH₃SO₃ + HO₂ is at least qualitatively in line with Ye et al. (2022), comparing their findings for high- and low-NO (for high-NO the HO₂ level should be quite small). Ye et al. (2022) already stated that

reactions of CH_3SO_3 with DMS or HCHO could lead to MSA. This could be also true for HONO, what they used in high-NO experiments as OH precursor. Here, more detailed investigations are needed. From our results, we can conclude, that a direct NO-mediated path to MSA do not exist, see Supplementary Figure 6 with a constant H_2SO_4 / MSA ratio with rising NO for a given CH_3SH or DMDS concentration.

We did no changes in the manuscript.

R2C9: In summary, I think this paper presents an advancement in organic reduced sulfur compound oxidation chemistry and I would like to see it published. My main request is that the authors clarify why their modeling results, especially the summary presented in Figure 4b, include so little H_2SO_4 produced via $\text{DMS} \rightarrow \text{HPMTF} \rightarrow \text{SO}_4^{2-}$, and why their results differ in the relevance of the abstraction pathway compared to modeling by others (Novak et al. 2021, Fung et al. 2022, Ye et al. 2022). I think it would also be wise to at least mention relevant sulfur compounds and chemistry that has been oversimplified in the text, such as SO_2 oxidation proceeding via many pathways (OH, O_3 , H_2O_2 , HOBr, HOCl, ISOPROOH, TMI-O₂) and HPMTF as a significant intermediate compound in organic reduced sulfur compound oxidation.

We thank the reviewer for the hint. In the new manuscript, we have included a broader description of the included total S(VI) formation pathways in CAPRAM. However, a deep discussion on this topic is not performed, because of the above-mentioned reasons that we do not focus on total sulfur (VI) formation, but on the gas-phase formation of H_2SO_4 (see our previous answers above).

"This mechanism system describes the formation of gas-phase H_2SO_4 and aqueous sulfate in a very detailed manner."

"With these two additional modules, CAPRAM4.0 includes almost all known sulfate formation pathways in the atmospheric aqueous phase, such as sulfite oxidation by H_2O_2 , O_3 , HNO_3 , reactive halogen species (X_2 radical or HOX, with X = Cl, Br or I) or transition metal ions."

Reviewer #3 (Remarks to the Author):

The authors report the observation of direct sulfuric acid formation from the oxidation of organic reduced-sulfur compounds which are normally emitted globally from natural sources over the ocean. They provide experimental evidence and implement newly observed reactions for model comparison to experimental findings. The sulfuric acid yield from these direct pathways were then adjusted to make it compatible with real atmospheric conditions. Finally, model simulations were performed to analyze the impact of this direct source of sulfuric acid from oceanic emission of reduced sulfur sources compared to traditional pathway of sulfuric acid formation (oxidation of OH+SO₂). The conclusions for importance of this direct route in the Southern hemisphere under pristine marine condition and the importance of considering loss routes via clouds chemistry is notable.

The paper is well written, well structured, and broadly covers some questions addressing knowledge gaps relevant to recent updates in marine sulfur chemistry the community has had since recent lab, field, and modeling work on novel atmospheric sulfur chemistry relevant to marine emission and their contribution for new particle formation. I would recommend publication after considering the minor points and suggestions below.

Thanks for the very positive feedback. Below, we provide a point-by-point answer to your comments.

R3C1: Line 68: While it is of course mentioned it in the main text, I think visually it would be good to include the formula of methylthiol, dimethyl sulfide and dimethyl disulfide in the figure caption along with the names, that way a reader can easily related the scheme and the figure caption.

Thanks for the good comment. We followed this suggestion and have extended the figure caption. According to reviewer 2 we are using now the acronyms DMS, DMDS and MSA and CH₃SH throughout in the manuscript.

"Fig. 1: Reaction scheme of the oxidation of reduced-sulfur emissions, i.e. CH₃SH (MeSH), CH₃SCH₃ (DMS) and CH₃SSCH₃ (DMDS). The scheme focuses on the reaction steps relevant for the formation of H₂SO₄ and MSA starting from CH₃S and CH₃SO₂ and is mainly based on Barnes *et al.*⁹. Signals of observed products in the present study are shown in bold. Dashed red arrows indicate complex reactions to the stated intermediates. Only the important main products of the individual pathways are displayed."

R3C2: Figure 4 — I suggest reversing the color bar of the legend for Figure 4(a) so that darker color represents higher values of average DMS concentration.

Based on the reviewer's suggestion, we have inverted the colour bar of Figure 5a so that darker colour represents higher average DMS mixing ratios. It should be noted that during the preparation of the manuscript we have tried already many different colour scales and had a

discussion among the authors about the most appropriate one. The created Figure with the inverted colour bar is shown below. Unfortunately, we think that the differences between individual measurement points are less clearly visible in the new Figure due to the logarithmic scaling. For example, the differences between the Pacific measurement sites are more apparent with the colour scale used previously. Therefore, the authors decided to keep the originally used colour scale.

Figure with the reversed colour scale as suggested by Reviewer#3.

R3C3: Supplementary Table 1 — Consider condensing the experiment description for column 1. Either rename the column or put the relative humidity value in a separate column. For column 4, I suggest renaming as 'Ionization technique' with entries as Nitrate/Iodide' for simplicity.

We did the needed changes in order to improve this table.

Supplementary Table 1: Experimental conditions of investigations described in the main text.

Experiment description	Reactants (molecules cm ⁻³)	Relative humidity (%)	OH source	Ionisation technique	Plot
Overview about product formation in the free jet flow system	[CH ₃ SH] = 6.6 x 10 ¹¹ [IPN] = (2.0 - 20) x 10 ¹¹	10	IPN photolysis	iodide	Fig. 2
Impact of CH ₃ SH on the formation of H ₂ SO ₄ and MSA, free jet flow system	[CH ₃ SH] = (6.7 - 202) x 10 ¹⁰ [IPN] = 4.0 x 10 ¹¹ [NO] = 1.0 x 10 ¹⁰	10	IPN photolysis	nitrate	Fig. 3
Impact of ozone on the formation of H ₂ SO ₄ or SO ₃ (dry conditions), laminar flow tube (LFT)	[CH ₃ SH] = 2.0 x 10 ¹¹ [CH ₃ SSCH ₃] = 6.1 x 10 ¹⁰ [TME] = 5.0 x 10 ⁹ [O ₃] = (1.5 - 13) x 10 ¹¹	10 or < 0.1	O ₃ + TME	nitrate	Fig. 4a
Impact of RO ₂ radicals on the formation of H ₂ Sat and MSA, LFT	[CH ₃ SH] = 2.0 x 10 ¹¹ [O ₃] = 5.7 x 10 ¹¹ [TME] = 0.5, 5.0, or 10 x 10 ¹⁰ [CH ₄] = 0, 1.0 or 2.0 x 10 ¹⁶	10	O ₃ + TME	nitrate	Fig. 4b
Impact of NO on H ₂ Sat formation, free-jet flow system or LFT	[CH ₃ SH] = 2.0 x 10 ¹¹ [CH ₃ SSCH ₃] = 6.1 x 10 ¹⁰ [TME] = 5.0, 15 or 50 x 10 ⁹ [O ₃] = 5.7 X 10 ¹¹ [NO] = (7.8 - 1000) x 10 ⁸	10	O ₃ + TME	nitrate	Fig. 4c
Impact of NO ₂ on H ₂ SO ₄ formation, free-jet flow system or LFT	[CH ₃ SH] = 2.0 x 10 ¹¹ [CH ₃ SSCH ₃] = 6.1 x 10 ¹⁰ [TME] = 5.0, 15 or 50 x 10 ⁹ [O ₃] = 5.7 X 10 ¹¹ [NO ₂] = (3.3 - 100) x 10 ⁹	10	O ₃ + TME	nitrate	Fig. 4d

R3C4:

- A) Supplementary Table 3 — Instead of just mentioning 'Averaged net reaction rates of daytime DMS, CH₃SH and SO₂ oxidation (in molecules cm⁻³ s⁻¹) over the period between the second to the fourth model day of the different simulations (in red)' in the figure caption add 'Day 2-4 average rate' for the red values in the body of the table.
- B) The manuscript does have lots of figures and tables and most of them have very descriptive titles, so more clarification/specification in the organization can enhance the clarity to the readers.

A) Following the reviewer's suggestion, we have revised Supplementary Table 3 (see below). We hope that the additional information in the body of the Table increases the readability of the Table. It should be noted that we have included 2 more simulation cases following suggestions of Reviewer#1. So, the revised Supplementary Table 3 looks as follows:

B) The authors thank the reviewer for the comment. We have particularly revised the Table captions to improve the clarity about the content given. Please see the manuscript with tracked changes for details.

Supplementary Table 3: Day 2-4 averaged rates of daytime DMS, CH₃SH and SO₂ oxidation (red) and primary daytime production rates of CH₃S and CH₃S₂ (blue) of the different simulations. The **bold** values refer towards the molar yields from DMS or CH₃SH oxidation, respectively. Normalized reaction rates (k_{int}) are calculated from the averaged net reaction rates divided by the modeled concentrations of DMS, CH₃SH and SO₂, respectively.

	DMS Oxidation		CH ₃ SH Oxidation	OH + SO ₂
	Addition/ molec. cm ⁻³ s ⁻¹	Abstraction/ molec. cm ⁻³ s ⁻¹	molec. cm⁻³ s⁻¹	molec. cm ⁻³ s ⁻¹
Cloud				
Total daytime 2-4 average rate (avg. kiss)	4.8x10 ⁴ (7.9x10 ⁴ s ⁻¹)	1.9x10 ⁴ (3.3x10 ⁴ s ⁻¹)	8.0x10 ³ (1.6x10 ⁻⁵ s ⁻¹)	7.0x10 ¹ (1.6x10 ⁻⁶)
Daytime 2-4 average rate of primarily formed CH ₃ S		4.1x10 ³ 22%	8.0x10 ³ 100%	
Daytime 2-4 average rate of primarily formed CH ₃ S ₂	2.1x10 ³ 4%			
Cloud with ten times higher NO emission rate				

Total daytime 2-4 average rate (avg. kiss)	4.8x10 ⁴ (8.8x 10 ⁶ s ⁻¹)	2.2x10 ⁴ (4.1x10 ⁷ s ^A)	8.2x10 ³ (1.8x10 ⁷ s ^A)	9.2x10 ¹ (2.1x10 ⁸ s ⁻¹)
Daytime 2-4 average rate of primarily formed CH ₃ S		4.1x10 ³ 19%	8.2x10 ³ 100%	
Daytime 2-4 average rate of primarily formed CH ₃ SO ₂	2.4x10 ³ 5%			

Cloud, lower HA (HA, 298K from Wollesen de Jonge et al., 2021)¹

Total daytime 2-4 average rate (avg. kiss)	4.8x10 ⁴ (7.9x10 ⁻⁶ s ⁻¹)	1.9x10 ⁴ (3.3x10 ⁻⁶ s ^A)	8.0x10 ³ (1.6x10 ⁻⁵)	7.4x10 ¹ (1.2x10 ⁻⁶ s ^A)
Daytime 2-4 average rate of primarily formed CH ₃ S		4.2x10 ³ 22%	8.0x10 ³ 100%	
Daytime 2-4 average rate of primarily formed CH ₃ SO ₂	7.3x10 ³ 15%			

no Cloud

Total daytime 2-4 average rate (avg. kiss)	5.1x10 ⁴ (9.2x10 ⁻⁶ s ⁻¹)	2.5x10 ⁴ (4.5x10 ⁻⁶ S ⁻¹)	8.5x10 ³ (2.1x10⁻⁵ S⁻¹)	1.2x10 ³ (4.6x10 ⁻⁷ s ¹)
Daytime 2-4 average rate of primarily formed CH ₃ S		6.8x10 ³ 27%	8.5x10 ³ 100%	
Daytime 2-4 average rate of primarily formed CH ₃ SO ₂	4.6x10 ³ 9%			

no Cloud with ten times higher NO emission rate

Total daytime 2-4 average rate (avg. kilt)	4.8x10 ⁴ (9.7x10 ⁶ s ⁻¹)	2.8x10 ⁴ (5.7x10 ⁶ s ⁻¹)	8.5x10 ³ (2.1x10 ⁵ S ⁻¹)	1.4x10 ³ (5.0x10 ⁻⁷ s ¹)
Daytime 2-4 average rate of		8.7x10 ³ 31%	8.5x10 ³ 100%	

primarily formed
CH₃S

Daytime 2-4
average rate of
primarily formed
CH₃SO₂ 4.7×10^3
10%

no Cloud, lower H_A (H_A, 298K from Wollesen de Jonge et al., 2021)¹

Total daytime 2-4
average rate (avg. kiss) 5.1×10^4
($9.0 \times 10^{-6} \text{ s}^{-1}$) 2.4×10^4
($4.4 \times 10^{-6} \text{ s}^{-1}$) 8.4×10^3
($2.1 \times 10^{-5} \text{ s}^{-1}$) 1.3×10^3
($4.4 \times 10^{-7} \text{ s}^{-1}$)

Daytime 2-4
average rate of
primarily formed
CH₃S 6.6×10^3
28% 8.4×10^3
100%

Daytime 2-4
average rate of
primarily formed
CH₃SO₂ 2.7×10^4
53%

R3C5:

- A) Supplementary Table 5 — Formation of HOOCH₂SCHO is considered as an important step for atmospheric DMS oxidation from recent observation and experimental studies. For CAPRAM-DM1.0, I find that photolysis of HOOCH₂SCHO was considered. But how does chemical loss of HOOCH₂SCHO by OH or it's cloud uptake do you think would contribute to the major findings of this present work?
- B) Unless I missed something to correlate, despite of ignoring reactions due to uncertainty in reaction kinetics, it would be great if it's possible to include a range of expected outcomes for such processes.

A) In the model, we also included reaction of HPMTF with OH according to the values given by Wu et al. (2015). Unfortunately, we forgot to include this in the provided mechanism table. We apologize for this and thank the reviewer for the excellent comment. We have gone through both Supplementary Table 5 and the chemical code applied to find missing reactions and included them. The revised Supplementary Table 5 is shown below:

Supplementary Table 5: Overview on the new implemented (termed new) or updated reactions in CAPRAM-DM1.0 in both, gas and aqueous phase. Second order rate coefficients are given in $\text{cm}^3 \text{ molecules}^{-1} \text{ s}^{-1}$ or $1 \text{ mol}^{-1} \text{ s}^{-1}$ for gas or aqueous phase, respectively, and first order rate coefficients in s^{-1} .

No.	Reaction	Rate coefficient at —295 K if not otherwise stated	Comment
Gas-phase reactions			

gD52	CH ₃ S(O)OH + OH → CH ₃ SO ₂ + H ₂ O	9.0x10 ⁻¹¹	forming solely CH ₃ SO ₂
new	CH ₃ SCH ₂ O ₂ →, HOOCH ₂ SCH ₂ O ₂	2.74x 10 ⁷ exp(-5950/T)	Ref. ¹⁶
new	HOOCH ₂ SCH ₂ O ₂ + HO ₂ → HOOCH ₂ SCH ₂ OOH + O ₂	1.91x 10 ⁻¹³ exp(1300/T)	MCMv3.3.1 ¹⁷
new	HOOCH ₂ SCH ₂ O ₂ + NO →, HOOCH ₂ SCH ₂ O + NO ₂	4.90x 10 ⁻¹² exp(260/T)	MCMv3.3.1 ¹⁷
new	HOOCH ₂ SCH ₂ O ₂ + NO ₃ →, HOOCH ₂ SCH ₂ O + NO ₂	2.30x 10 ⁻¹²	MCMv3.3.1 ¹⁷
new	HOOCH ₂ SCH ₂ O ₂ →, 0.8 HOOCH ₂ SCH ₂ O + 0.1 HOOCH ₂ SCH ₂ OH + 0.1 HOOCH ₂ SCHO	3.74x 10 ⁻¹²	MCMv3.3.1 ¹⁷
new	HOOCH ₂ SCH ₂ O ₂ →, HOOCH ₂ SCHO + OH	4.20x 10 ⁷ exp(-5390/T)	Ref. ¹⁶
new	HOOCH ₂ SCH ₂ O + O ₂ → HOOCH ₂ SCHO + HO ₂	2.50x 10 ⁻¹⁴ exp(-300/T)	est. MCMv3.3.1 ¹⁷
new	HOOCH ₂ SCH ₂ O → HOOCH ₂ S + HCHO	1.00x 10 ⁶	Ref.'s
new	HOOCH ₂ SCH ₂ OH + OH →, HOOCH ₂ SCHO + HO ₂	2.78x 10 ⁻¹¹	as for CH ₃ SCH ₂ OH in MCMv3.3.1 ¹⁷
new	HOOCH ₂ SCH ₂ OOH + hv → HOOCH ₂ SCH ₂ O	J=1.53 x 10 ⁻⁵ cos(φ) ^{0.682} exp(-0.279 / cos(φ))	as for CH ₃ OOH
new	HOOCH ₂ SCH ₂ OOH + OH → HOOCH ₂ SCH ₂ O ₂ + H ₂ O	7.63x 10 ⁻¹³ exp(635/T)	Ref. ¹⁹
new	HOOCH ₂ SCH ₂ OOH + OH →, HOOCH ₂ SCHO + OH + H ₂ O	7.03x 10 ⁻¹¹	as for CH ₃ SCH ₂ OOH in MCMv3.3.1 ¹⁷
new	HOOCH ₂ SCHO + hv →, HCHO + OCS + HO ₂ + OH	J=7.649x 10 ⁻⁶ cos(φ) ^{0.682} exp(-0.338 / cos(φ))	as for CH ₃ OOH in MCMv3.3.1 ¹⁷
new	HOOCH ₂ SCHO + hv → HOOCH ₂ S + CO + HO ₂	J=2.792x 10 ⁻⁵ cos(φ) ^{0.682} exp(-0.279 / cos(φ))	as for CH ₃ SCHO in MCMv3.3.1 ¹⁷
new	HOOCH ₂ SCHO + OH →, HOOCH ₂ SCO	1.40x 10 ⁻¹²	Ref. ¹⁵
new	HOOCH ₂ SCO → OH + HCHO + OCS	9.20x 10 ⁹ exp(-505.4/T)	Ref.'s
new	HOOCH ₂ SCO → HOOCH ₂ S + CO	1.60 x 10 ⁹ exp(-1468.6/T)	Ref. ¹⁸
new	HOOCH ₂ S + O ₃ → HOOCH ₂ SO + O ₂	1.15 x 10 ⁻¹² exp(430/T)	Ref.'s
new	HOOCH ₂ S + NO ₂ → HOOCH ₂ SO + NO	6.00x 10 ⁻¹¹ exp(240/T)	Ref.'s
new	HOOCH ₂ SO + O ₃ → HCHO + OH + O ₂	4.00x 10 ⁻¹³	Ref. ¹⁸
new	HOOCH ₂ SO + NO ₂ → SO ₂ + HCHO + OH + NO	1.20x 10 ⁻¹¹	Ref. ¹⁵

new	CH ₃ SH + OH ⁻ , CH ₃ S + H ₂ O	$9.9 \times 10^{-12} \exp(356/T)$	Ref. ⁴
new	CH ₃ SH + NO ₃ -> CH ₃ S + HNO ₃	9.2×10^{-13}	Ref. ⁴
new	CH ₃ SH + Cl -> CH ₃ S + HCl	$1.2 \times 10^{-10} \exp(150/T)$	Ref. ⁴
new	CH ₃ SH + BrO ⁻ , CH ₃ S + HOBr	$2.2 \times 10^{-15} \exp(827/T)$	Ref. ²⁰
Aqueous-phase reaction			
new	CH ₃ S(O)(Br)CH ₃ + H ₂ O -> CH ₃ S(O)OH + HBr + CH ₃	1.0×10^2	est. as for CH ₃ S(O)(CDCH ₃) ²¹

B) As outlined to reviewer#2, the oxidation of HPMTF in the gas-phase oxidation contributes about 37% and 33% to the gas-phase SO₂ formation in the simulation AP Cloud and AP Cloud with lower HA, respectively. However, it is important to note that the HPMTF concentration is larger than 50 ppt at the end of all simulations. This is a bit in discrepancy to the measured values of Veres et al. (2020), who state (in their correction) that the HPMTF "mixing ratios frequently exceeding 10 ppt in the Marine Boundary Layer (MBL) and periodically as large as 100 ppt". The discrepancy between modelled and measured values is possibly due to the not included uptake and further oxidation of HPMTF in the tropospheric aqueous phase. The Henry's Law coefficient of HPMTF is calculated to be $> 10^4 \text{ mol L}^{-1} \text{ atm}^{-1}$ (Wollesen de Jonge et al., 2021). This value means that during cloud passage of the air parcel all HPMTF is taken up into cloud droplets, where it could rapidly react. Moreover, the solubility of hydrated HPMTF was found to similar to MSA (Wollesen de Jonge et al., 2021) meaning that an effective uptake of HPMTF even on deliquesced aerosol particles could occur, where it might also rapidly react. Therefore, a contribution to gas-phase SO₂ could be even smaller. As the paper focuses on direct gas-phase H₂SO₄ formation and the findings of the present study are not affected by the current uncertainties in the HPMTF chemistry, we omit a deeper discussion of the uncertainties. The uncertainty issue is now mentioned in the main manuscript.

The extended text in the revised manuscript reads as follows:

"The mechanistic updates comprise the formation of the hydroperoxymethyl thioformate (HPMTF) and its further oxidation in the gas phase. The gas-phase HPMTF oxidation follows mainly the proposed routes described by Wu et al. (2015)¹¹, considering SO₂ or OCS formation, only. Thus, HPMTF cannot contribute to the direct gas-phase formation of H₂SO₄ in this mechanism. Phase transfer and subsequent aqueous-phase processing of HPMTF, not included yet because of the current high uncertainties, do not change this."

R3C6: General comment: Is it possible to evaluate the aerosol number concentrations with this additional source of direct sulfuric acid formation using the model used in this study.

(AT)

In the applied box model SPACCIM (Spectral Aerosol Cloud Chemistry Interaction Model; Wolke et al., 2005), aerosol particle growth occurs through condensation of gas-phase compounds and secondarily through aqueous-phase chemistry processes leading to the

formation less-volatile compounds which remain on the particle increase the particulate matter. Unfortunately, SPACCIM is not designed to deal new particle formation to investigate the resulting effect of the direct gas-phase H₂SO₄ formation. We addressed this in the methods part, now.

"It should be noted that, the applied model is not designed to deal new particle formation to investigate the resulting effect of the direct gas-phase H₂SO₄ formation. Therefore, we only focus on the chemical gas-phase H₂SO₄ formation in the present study."

Regarding the "additional source", we would not call it "additional". The observed ambient NPF rates include this pathway already but the current models do not. Based on the obtained relative contribution of the direct gas-phase H₂SO₄ formation pathway, the potential contribution to the real NPF rates can be estimated from the findings of the present study. Moreover, the result of the present study implies that, considering the different diurnal concentration pattern of DMS, CH₃SH and SO₂ (gas-phase H₂SO₄ precursors) often observed in ambient marine air masses (see e.g. Gray et al. (2011), Davis et al. (1999), and Novak et al. (2022)), the relative contribution of the direct gas-phase H₂SO₄ formation to NPF can vary during the day and might be higher in the forenoon. This finding can help to understand NPF in clean pristine regions which is still controversy discussed (see Kerminen et al. (2018) and references therein).

Literature

- Berndt, T., Scholz, W., Mentler, B., Fischer, L., Hoffmann, E. H., Tilgner, A., Hyttinen, N., Prisle, N. L., Hansel, A., Herrmann, H. (2019), Fast peroxy radical isomerization and OH recycling in the reaction of OH radicals with dimethyl sulphide. *J. Phys. Chem. Lett.*, 10, 6478-6483. <https://doi.org/10.1021/acs.jpcclett.9b02567>
- Davis, D., Chen, G., Bandy, A., Thornton, D., Eisele, F., Mauldin, L., Tanner, D., Lenschow, D., Fuelberg, H., Huebert, B., Heath, J., Clarke, A., Blake, D. (1999) Dimethyl sulfide oxidation in the equatorial Pacific: Comparison of model simulations with field observations for DMS, SO₂, H₂SO₄(g), MSA(g), MS and NSS, *J. Geophys. Res.- Atmos.*, 104, 5765-5784. <https://doi.org/10.1029/1998jd100002>.
- Gray, B. A., Wang, Y., Gu, D., Bandy, A., Mauldin, L., Clarke, A., Alexander, B., Davis, D. D. (2010), Sources, transport, and sinks of SO₂ over the equatorial Pacific during the Pacific Atmospheric Sulfur Experiment, *J. Atmos. Chem.*, 68, 27-53. <https://doi.org/10.1007/s10874-010-9177-7>.
- Kerminen, V.-M., Chen, X., Vakkari, V., Pethja, T., Kulmala, M., Bianchi, F. (2018), Atmospheric new particle formation and growth: review of field observations, *Environ. Res. Lett.*, 13, 103003. <https://doi.org/10.1088/1748-9326/aadf3c>.
- Jernigan, C.M., Cappa, C. D., Bertram, T. H. (2022), Reactive Uptake of Hydroperoxymethyl Thioformate to Sodium Chloride and Sodium Iodide Aerosol Particles. *J. Phys. Chem. A* 126 (27), 4476-4481. <https://doi.org/10.1021/acs.jpca.2c03222>.
- Novak, G. A., Kilgour, D. B., Jernigan, C. M., Vermeuel, M. P., Bertram, T. H. (2022), Oceanic emissions of dimethyl sulfide and methanethiol and their contribution to sulfur dioxide production in the marine atmosphere, *Atmos. Chem Phys.*, 22, 6309-6325. <https://doi.org/10.5194/acp-22-6309-2022>.
- Reed, C., Evans, M. J., Crilley, L. R., Bloss, W. J., Sherwen, T., Read, K. A., et al. (2017), Evidence for renoxification in the tropical marine boundary layer, *Atmospheric Chemistry and Physics*, 17(6), 4081-4092, <https://doi.org/10.5194/acp-17-4081-2017>.
- Tilgner, A., Schaefer, T., Alexander, B., Barth, M., Collett Jr., J. L., Fahey, K. M., Nenes, A., Pye, H. O. T., Herrmann, H., McNeill, V. F. (2021), Acidity and the multiphase chemistry of atmospheric aqueous particles and clouds. *Atmos. Chem. Phys.*, 21, 13483-13536. <https://doi.org/10.5194/acp-21-13483-2021>.
- Veres, P. R., Neuman, J. A., Bertram, T. H., Assaf, E., Wolfe, G. M., Williamson, C. J., Weinzierl, B., Tilmes, S., Thompson, C. R., Thames, A. B., Schroder, J. C., Saiz-Lopez, A., Rollins, A. W., Roberts, J. M., Price, D., Peischl, J., Nault, B. A., Moller, K. H., Miller, D. O., Meinardi, S., Ryerson, T. B. (2020). Global airborne sampling reveals a previously unobserved dimethyl sulfide oxidation mechanism in the marine atmosphere. *Proc. Natl. Acad. Sci. USA*, 117(9), 4505-4510. <https://doi.org/10.1073/pnas.1919344117>.
- Wolke, R., Sehili, A. M., Simmel, M., Knoth, O., Tilgner, A., & Herrmann, H. (2005), SPACCIM: A parcel model with detailed microphysics and complex multiphase chemistry, *Atmo. Environ.*, 39(23/24), 4375-4388, <https://doi.org/10.1016/j.atmosenv.2005.02.038>.
- Wollesen de Jonge, R., Elm, J., Rosati, B., Christiansen, S., Hyttinen, N., Liidemann, D., Bilde, M., and Roldin, P. (2021), Secondary aerosol formation from dimethyl sulfide — improved mechanistic understanding based on smog chamber experiments and modelling. *Atmos. Chem. Phys.*, 21, 9955-9976, <https://doi.org/10.5194/acp-21-9955-2021>.
- Wu, R.; Wang, S.; Wang, L. New mechanism for the atmospheric oxidation of dimethyl sulfide. The importance of intramolecular hydrogen shift in a CH₃SCH₂OO radical. *J. Phys. Chem. A*, 2015, 119, 112-117. <https://doi.org/10.1021/jp511616j>.
- Ye, Q., Goss, M. B., Krechmer, J. E., Majluf, F., Zaytsev, A., Li, Y., Roscioli, J. R., Canagaratna, M., Keutsch, F. N., Heald, C. L., Kroll, J. H. (2022), Product distribution, kinetics, and aerosol formation from the OH oxidation of dimethyl sulfide under different RO₂ regimes, *Atmos. Chem. Phys.*, 22, 16003-16015, <https://doi.org/10.5194/acp-22-16003-2022>.

REVIEWERS' COMMENTS

Reviewer #1 (Remarks to the Author):

The authors had revised their manuscript carefully, now it is acceptable as it is.

Reviewer #2 (Remarks to the Author):

I appreciate the authors' extensive revisions and thoughtful responses to my comments. The manuscript is much improved by clarifying the writing throughout, including by changing "direct" to "gas-phase" when describing the goal of the study in the title, abstract, and elsewhere. Changing the names of compounds (e.g., DMDS, MSA) will make it easier for a broad audience to read. The writing is also improved by adding more detailed explanations and subheadings in the results and discussion sections. Overall, the authors have adequately addressed my concerns and this study will be of broad interest and relevance to the communities studying sulfur, modelling, new particle formation, and atmospheric chemistry.